# Control theory illustrates the energy efficiency in the dynamic reconfiguration of functional connectivity

Shikuang Deng[1], Jingwei Li [2,3], B. T. Thomas Yeo[4,5,6,7,8] & Shi Gu [1✉]

The brain's functional connectivity fluctuates over time instead of remaining steady in a stationary mode even during the resting state. This fluctuation establishes the dynamical functional connectivity that transitions in a non-random order between multiple modes. Yet it remains unexplored how the transition facilitates the entire brain network as a dynamical system and what utility this mechanism for dynamic reconfiguration can bring over the widely used graph theoretical measurements. To address these questions, we propose to conduct an energetic analysis of functional brain networks using resting-state fMRI and behavioral measurements from the Human Connectome Project. Through comparing the state transition energy under distinct adjacent matrices, we justify that dynamic functional connectivity leads to 60% less energy cost to support the resting state dynamics than static connectivity when driving the transition through default mode network. Moreover, we demonstrate that combining graph theoretical measurements and our energy-based control measurements as the feature vector can provide complementary prediction power for the behavioral scores (Combination vs. Control: $t = 9.41$, $p = 1.64e{-}13$; Combination vs. Graph: $t = 4.92$, $p = 3.81e{-}6$). Our approach integrates statistical inference and dynamical system inspection towards understanding brain networks.

[1] School of Computer Science and Engineering, University of Electronic Science and Technology of China, Chengdu, China. [2] Institute of Neuroscience and Medicine, Brain & Behaviour (INM-7), Research Center Jülich, Jülich, Germany. [3] Institute for Systems Neuroscience, Medical Faculty, Heinrich-Heine University Düsseldorf, Düsseldorf, Germany. [4] Department of Electrical and Computer Engineering, National University of Singapore, Singapore 117583, Singapore. [5] Centre for Sleep & Cognition & Centre for Translational Magnetic Resonance Research, Yong Loo Lin School of Medicine, Singapore, Singapore. [6] N.1 Institute for Health & Institute for Digital Medicine, National University of Singapore, Singapore, Singapore. [7] Integrative Sciences and Engineering Programme (ISEP), National University of Singapore, Singapore, Singapore. [8] Athinoula A. Martinos Center for Biomedical Imaging, Massachusetts General Hospital, Charlestown, MA 02129, USA. ✉email: gus@uestc.edu.cn

Large-scale noninvasive neuroimaging like functional magnetic resonance imaging (fMRI) provides an accessible window into the complex neurophysiological dynamics of the human brain. Such dynamics are supported by a relatively fixed backbone of white matter fiber bundles spanning cortical and subcortical structures in an intricate network characterized by highly nontrivial topology[1,2]. Although the underlying white matter fibers are static at least within hours, the functional dynamics of the whole system are highly nonlinear and extremely difficult to identify. One approach is to simplify the dynamics with approximate linear models[3,4] or dynamical mean-filed models[5] under the constraint of underlying white matter network architecture. As an alternative, another group of seminal works investigate the large-scale functional dynamics with functional connectivity that are defined as certain types of statistical association between the observed signal sequences for different brain areas[6,7]. These two families of approaches constitute the dynamical and statistical paths towards the understanding of brain network.

While these works provide insightful description of the functional connectome separately from the statistical and mechanistic views, the isolated studies on statistically defined connectivity and the rich dynamics of functional neuroimaging leave it vacant to mechanistically explain the principle of dynamic reconfiguration[8,9]. Especially, it is not well understood how the specialized reconfiguration of the brain's functional network leads to high network efficiency that mechanistically benefits neural signal processing. Furthermore, considering that the brain activity involves both the metabolic and signal-spreading procedure, this integration of the statistical and mechanistic views may generate novel network-based biomarkers that improve the predictive power for individual differences in human behaviors[10–12]. Thus, it is of critical significance to build a dynamical model that can illustrate how the reconfiguration of the brain's functional networks can facilitate the state transition in the brain. Furthermore, the proposed dynamic model should be able to derive effective biomarkers with complementary power to traditional graph-theoretical measurements when predicting human behavioral scores[13].

Network control theory is a particularly promising mathematical framework to model the structure-induced brain state reconfiguration[14,15]. Originally developed in the physics and engineering literature[16], the approach is flexible to applications across scales and species, including cellular models[17], *C. elegans*[18], fly[19], mouse[19], macaque[20], and human[20], and has been extended to study cognitive function[21–23], development[24], heritability[25], disease[26,27], and the effects of stimulation[28–32]. Such broadening utility of network control theory in neuroscience is built on the empirical estimate of the network's structural architecture and the modeling of the dynamics that such an architecture can support[19,20]. On the one hand, the dependence on structural connectome brings convenience in characterizing the evolving whole-brain states through analytical tools. On the other hand, current works utilizing notions of network control are to some degree limited by the assumption that all effective relations between regions are time-invariant and encapsulated in the underlying white matter network architecture. This assumption leaves the approach agnostic to distinct ways in which the connection diagram can be utilized for inter-regional communication[33], both to support diverse states in health[21] and disease[34].

In the practice of applying network control theory, the dynamics of the examined system are highly determined by the interaction matrix that describes how the system evolves from the current state to the future state. Thus, the control model's sensitivity to the functional connectivity elicited by a given brain state can be partially attained by constructing control model with interaction matrix induced from functional time series. Given the time-varying nature of brain dynamics[8], both static and dynamic functional connectivity need to be considered when we conceptualize brain state transitions as control trajectories induced by distinct interaction diagram. Static functional connectivity measures the synchronization or asynchronization of the fMRI time series across brain regions. When the static functional connectivity matrix is used as the interaction matrix in the control model, the same intuition applies as in the case for structural connectivity[35], where higher link weight suggests stronger diffusive signal progression along the links[20]. Complementary to the static functional connectivity, dynamic functional connectivity focuses on the time-varying aspect of brain dynamics. The functional dynamics of brain networks can be captured by dynamic causal model[36], structural equation model[37], auto-regressive model[38], Granger causality and transfer entropy[39] or a series of connectivity matrices called dynamic functional connectivity[7]. Recent works suggest that dynamic functional connectivity can predict scores in behavioral tasks complementary to static connectivity[11]. Applying control theories on both static and dynamic connectivity might potentially provide mechanistic explanations on specific patterns of how functional organization fluctuates. Meanwhile, this approach might also help to design powerful biomarkers to predict individual differences in cognitive performance.

Here we use fMRI data and behavioral measurements[11] from the Human Connectome Project Young Adult S1200 release[40,41]. First, we build the canonical linear time-variant model to characterize dynamics of brain's state transition. Next, we vary the interaction matrix, the initial state, and the target state in the model to estimate the data-driven functional controllability measurements[20] and the energy required to drive the corresponding optimal trajectories across states[22]. Based on this model of controllability for both the static and dynamic functional connectivity, we test two specific hypotheses. *Hypothesis I*: The modeled brain system with temporal interaction matrices estimated as the series of dynamic functional connectivity is energetically more efficient than those with static interaction matrix and with temporally random shuffled dynamic functional connectivity. *Hypothesis II*: The energy-based control theoretical biomarkers can predict the behavioral scores complementary to the traditional network nodal measurements including the weighted degree, local efficiency and participation coefficient considering its specialty in characterizing the transition energy.

## Results

We use 865 subjects in the HCP Young Adults 1200-subjects release resting-state functional fMRI data[40,42], include four sessions of $419 \times 1200$ (Region × TR) time series including 400 cortical areas[43] and 19 subcortical areas[44] (Fig. 1a). In general, the main results focus on the first session and the remaining sessions are used to investigate the reproducibility of the main results in the supplement.

**Controllability induced by different types of connectivity.** In the brain control analysis framework, a state refers to a vector $\mathbf{x}(t)$ that encodes the neurophysiological activity map across the whole brain. In the current work, the $\mathbf{x}(t)$ is the BOLD activation map that evolves temporally during the resting states. In the general context of control theory, the evolutionary dynamics of the state $\mathbf{x}(t)$ is formulated as an equation relating the first-order derivative of the state $\mathbf{x}(t)$ to the state variable $\mathbf{x}$ itself and the control input. Specifically, in this work, we model the brain as a linear control system. For this system, given the initial and target states, the control trajectory

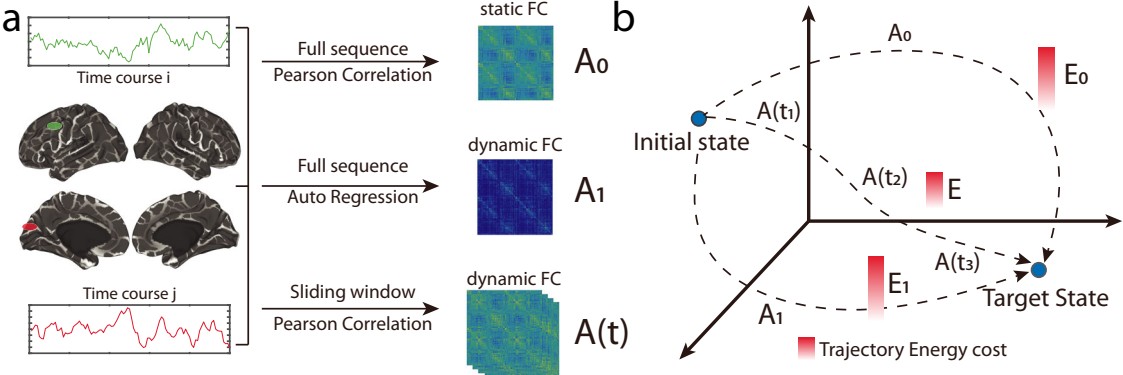

**Fig. 1 Conceptual schematic. a** We begin with the preprocessed BOLD time series from 400 cortical (Schaefer atlas[43]) and 19 subcortical regions. The static functional connectivity matrix is calculated as the Pearson's correlation across the full-time sequence. Two types of dynamic functional connectivity are estimated by the autoregression approach[38] and the sliding-window approach[6,85], respectively. **b** For a pair of given initial and target states, different types of interaction matrices would result in distinct trajectories thus varied energetic cost.

moving from the initial to target states is determined by the interaction matrix $\mathbf{A}(t)$, the input matrix $\mathbf{B}$, and the control input $\mathbf{u}(t)$, in the form of $\dot{x}(t) = \mathbf{A}(t) \cdot \mathbf{x}(t) + \mathbf{B} \cdot \mathbf{u}(t)$. The state interaction matrix characterizes how the control system moves from the current state to the future state. In the present study, it is defined based on either the static or the dynamic functional connectivity. The control input matrix $\mathbf{B}$ denotes the location of control nodes on which we place the input energy. The control input $\mathbf{u}(t)$ denotes the exact form of the control strategy. Different types of interaction matrices might induce different types of control dynamics that vary in the efficiency of energy usage. Here we investigate three types of control dynamics induced from (1) static functional connectivity via Pearson's correlation (sFC-Correlation), (2) dynamic functional connectivity via the autoregression (dFC-Autoregression), and (3) dynamic functional connectivity sequence via sliding-window Pearson's correlation (dFC-Slidingwindow). When the interaction matrix is set as the sFC-Correlation, each element in the matrix then indicates the coupling strength of BOLD activity between the two corresponding regions. Thus intuitively, we can interpret the induced control dynamics as the noise-driven hemodynamics[45]. When the interaction matrix is set as the dFC-Autoregression, the matrix element then denotes the temporal dependence of the signal series between regions, suggesting that the induced control dynamics characterizes the process of information propagation[46]. When the interaction matrix is set as the dFC-Slidingwindow thus time dependent, the dynamics can then be viewed as a hybrid result from both the noise-driven hemodynamics and information propagation. In our current framework, we assume that $\mathbf{u}(t)$ is determined through the optimization of control energy. Under this setup, we can then investigate the role of different regions in controlling the linear dynamics induced by distinct types of connectivity matrices via varying the regions included in the control set.

**Preference in control strategy.** Next, we iterate each region as the controller of the dynamics when the interaction matrices are modeled by the sFC-Correlation, dFC-Autoregression, and dFC-Slidingwindow matrices separately. Based on the constructed dynamics as well as the associated statistics, we obtain the distributions of average and modal controllability that quantify the volume of states the system can arrive and distance the system can reach given a unit of input energy, respectively. The 419 regions are grouped into 8 brain networks that are widely used in the works of investigating functional networks[43,47]. And the controllability when setting each region as the controller is averaged within each network.

Although it remains unknown how the brain intrinsically drives the transition across different states, we can utilize these two

measurements to infer each region's preference in control strategy following the basic rule that the brain must cost energy that depends on connectome and states for the dynamical reconfiguration. If the average controllability in a region is higher, it suggests that energy input on the region would be more efficient if the goal is to cover as more states as possible. Thus intuitively, it quantifies the regional ability in moving the whole brain to many easily reachable states. Respectively, the modal controllability gives a description of the regional ability in moving the whole brain to difficult-to-reach states, respectively[20]. Our findings unveil that overall the average controllability of the dynamics induced from dFC-Slidingwindow is higher than that of the sFC-Correlation and dFC-Autoregression (Fig. 2a) except for the subcortical areas. The modal controllability exhibits higher values when the control system is parameterized by dFC-Autoregression than that by the sFC-Correlation and dFC-Slidingwindow (Fig. 2b). And the student's $t$-test results of the system-wise average and modal controllability from three cases are included in Tables S1, S2. These results suggest that the brain's time-varying correlation networks are preferable to the average control strategy, i.e., driving the network into many easily reachable states. In contrast, the dFC-Autoregression is more preferable to the modal control strategy, i.e., moving the network into specific states, or in another words, task execution. The regions with high average controllability are located in the subcortical and limbic areas when the control dynamics are induced from the sFC-Correlation. When the control dynamics is induced from dFC-Autoregression or dFC-Slidingwindow dynamics, the front-parietal and default mode areas exhibit high values in the average controllability. This suggests that the default mode and fronto-parietal networks play a fundamental role in supporting the dynamic reconfiguration of connectome in the resting states by efficiently driving the state transition into many easily reachable states. In terms of the modal controllability across regions, the default mode and fronto-parietal areas display high values for the control dynamics with sFC-Correlation, while the subcortical and salience areas exhibit high values for dFC-Autoregression and dFC-Slidingwindow induced dynamics. This means that the subcortical and salience areas are more efficient in driving the state transition to the difficult-to-reach states, i.e., executing complex tasks, supporting the subcortical area's role as information hubs, and salience area's role as attentional control hubs of the nervous system[48]. The reproducibility results of the control theoretic measurements are included in Note S1 and Fig. S1.

**Energetic advantage of time-variant control paradigm.** In the former section, we show that dFC-Slidingwindow is more advantageous than sFC-Correlation from the perspective of

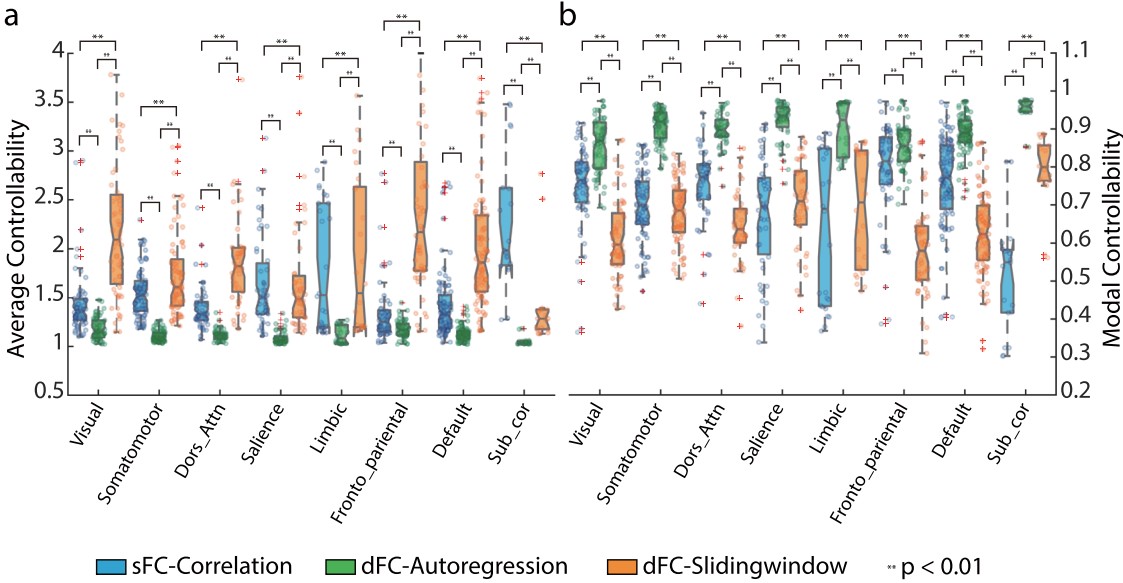

**Fig. 2 Spatial distribution of functional controllability.** We show the distribution of mean (**a**) average and (**b**) modal controllability across the different functional brain networks. The mean average control values (Mean ac-Value) and mean modal control values (Mean mc-Value) are calculated across all subjects. The system mean is calculated across the within-system regions for all subjects. We adopt the eight brain networks from Yeo's partition to compare differences in controllability for different networks[43,86].

average controllability. However, in the real brain state transition, only limited number of states can be reached and the control nodes in function can vary in different circumstances. Thus, to practically examine whether the specific pattern of dynamic functional connectivity is more advantageous in controlling the brain's resting-state transition, here we compare the energy cost for the state transitions driven by sFC-Correlation and dFC-Slidingwindow with the initial and target states sampled from the observed BOLD series. Specifically, we test the following two progressive sub-hypotheses in Hypothesis I. *Hypothesis I-a*: The control paradigm induced by dFC-Slidingwindow is energetically more efficient in driving the state transition than that induced by the sFC-Correlation. *Hypothesis I-b*: The control paradigm induced by dFC-Slidingwindow following the observed order is also advantageous in energy cost than the one induced by random dFC-Slidingwindow where the sequence of functional connectivity is random permuted across time-windows.

In terms of Hypothesis I-a, we can see from Fig. 3a that the temporal control paradigm is significantly more energetically efficient than the time-invariant control paradigm no matter whether we choose the control set as default mode network ($t = 46.22$, $p = 9.36e-236$, $n = 865$), the combination of default model (DM), dorsal attention (DA), salience (SA), and fronto-parietal networks (FP) ($t = 43.65$, $p = 9.92e-221$, $n = 865$), or the whole network ($t = 38.72$, $p = 5.38e-191$, $n = 865$). Next, we explore how the time interval $\tau$ in the control dynamics is related to the control energy as $\tau$ determines the decay of effect by the input energy in early time thus largely affects the overall energy cost[49]. We can see that when controlling on the default mode network, the improvement of energetic cost is higher on short control intervals than the long control intervals. Especially, the control energy of dFC-Slidingwindow has a minimum around $\tau \approx 5.5$ (Fig. 3b). Similar patterns exist when the control set is the combination of default mode, dorsal attention, salience, and fronto-parietal networks but with less significance for the energy difference (Fig. 3c, d). Note S6 and Table S3 provide the student's $t$-test results of the energy efficiency comparison (Fig. 3b–d) within different intervals of control time.

In terms of Hypothesis I-b, similar analysis is conducted to compare the energy cost corresponding to the observed order of dFC-Slidingwindow matrices with the null distribution built from the energy costs corresponding to the randomly permuted dynamic connectivity matrices. From Fig. 3e, we can see that compared to the random reconfiguration, the reconfiguration of connectivity matrices in observed order costs less energy no matter we set the control drivers as default mode network ($t = 22.95$, $p = 2.29e-91$, $n = 865$), the combination of default model, dorsal attention, and salience network ($t = 35.46$, $p = 1.04e-170$, $n = 865$), or the whole network ($t = 58.13$, $p = 8.15e-301$, $n = 865$). This supports that the order of reconfiguration makes a difference in addition to the benefit from temporal control paradigm. We further find that the relative ratio of the difference in log-energy cost is negatively correlated with the log-energy cost of the dynamics induced by the original dynamic functional connectivity. In Fig. 3f–h, the x-axis denotes the log of the energy required to make the transition and each point represent such a transition determined by the initial and target states. The trend then represents the relationship between the ratio of additional energy when the transition is driven by random matrix in sliding window to the energy calculated with the observed order of dynamic functional connectivity in sliding window. Thus, the decreasing trend indicates that the state transition that requires less energy to drive benefits more from following the observed order of dynamic connectivity reconfiguration. This indicates that the state transition that requires less energy to drive benefits more from following the observed order of dynamic connectivity reconfiguration. It suggests that the brain network adapts the strategy to be preferable to the easily reachable states to maintain the efficiency of overall energy utility. The results after changing the initial and final state to the BOLD value of the initial and final moments are included in the Note S6 and Fig. S7.

**Predict behavioral phenotypes from control theoretical measurements.** In the previous section, we show that control theory can explain the appearance of dynamic reconfiguration in the

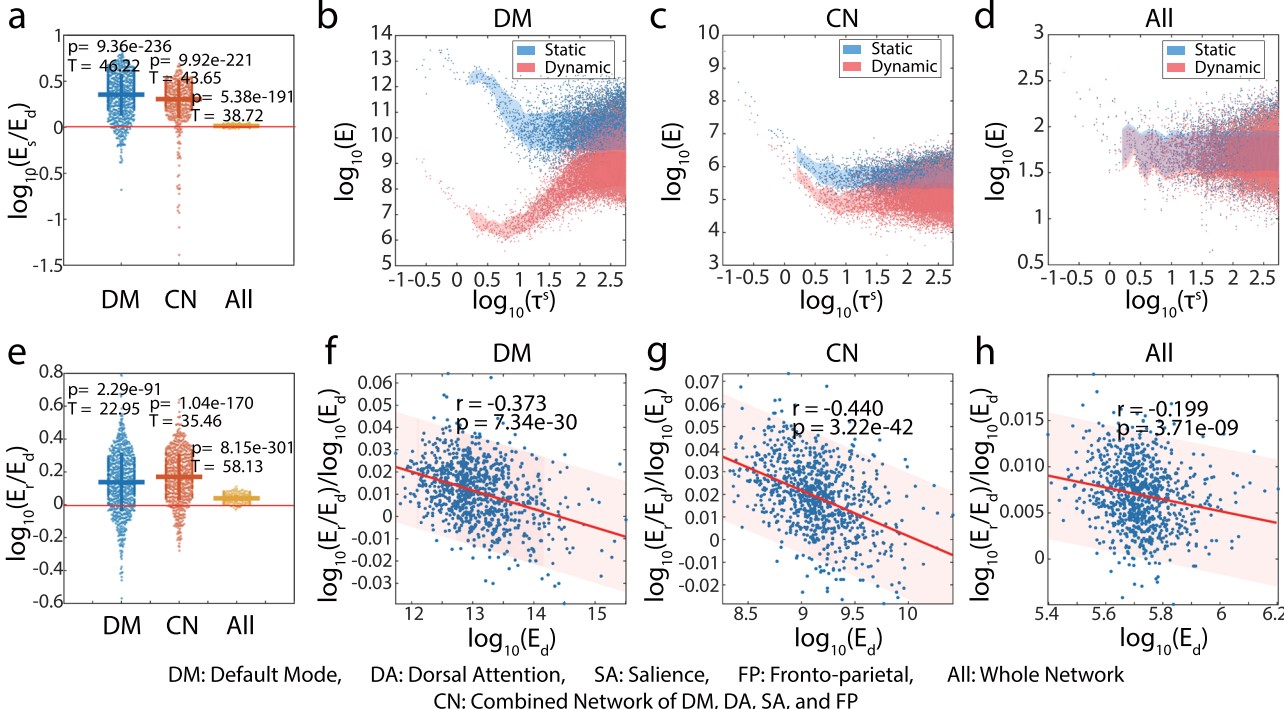

DM: Default Mode,   DA: Dorsal Attention,   SA: Salience,   FP: Fronto-parietal,   All: Whole Network
CN: Combined Network of DM, DA, SA, and FP

**Fig. 3 Energy efficiency explains the dynamic reconfiguration of functional connectivity.** Each point in (**a**–**h**) represents a transition between a pair of initial and target states. The $E_s$ is the energy cost when the control dynamics are induced by the sFC-Correlation. The $E_d$ is the energy cost when the control dynamics is induced by the dFC-Slidingwindow. The $E_r$ is the energy cost when the control dynamics is induced by the same matrices of the dFC-Slidingwindow but in a randomly shuffled order. The y-axis of (**f**)–(**h**) represents the log of the relative energy inflation when changing the connectivity matrices from the observed order to the random order. **a** We successively set Default Mode network (DM), the combination (CN) of Default Mode, Dorsal Attention (DA), Salience (SA), and Fronto-parietal networks (FP), or the whole network (All) as the control set. The T-tests measure their significance vs. zero. **b** When the network is controlled through DMN with varied control intervals ($\tau$), the improvement of energetic cost (i.e., the vertical difference between blue and red dots) is higher on short control intervals (i.e., smaller $\tau$) than the long control intervals. Similar patterns exist when **c** the control set is the combination of DMN, DA, SA, and FP but with less significance. **d** The difference is less visually significant when controlling on the whole network. **e** When we permute the order of sequential matrices in the temporal control paradigm, we also observe an increase in the energy cost. **f**–**h** The relative increase of log energy is negatively correlated to the log energy corresponding to the observed order of reconfiguration for dynamic functional connectivity.

resting state and the energy cost of state transition characterizes the network efficiency differently from path-based measurements in graph theory. Thus it becomes practically possible and meaningful to further ask whether the control measurements can facilitate the prediction of behavioral performance scores differently from the graph-theoretical measurements.

In this section, we first compare the predictive power of control and graph-theoretical features on the behavior task scores in HCP. For control theoretical measurements, we adopt the average controllability, modal controllability, and the regional activation energy that denotes the energy needed to activate a single region[22]. For graph-theoretical measurements, we take three most common regional measurements, weighted nodal degree, local efficiency, and participation coefficients[13]. Both the static and dynamic functional connectivity (autoregression) matrices are included to construct the input features for better prediction performance[11]. The full input feature of the predictive model is a $6N \times 1$ vector that concatenates the control or graph-theoretical features based on both the static and dynamic functional connectivity matrices. Here $N$ is the number of regions. We use the kernel ridge regression model[50] to obtain the predicted values via 10-fold cross-validation and compute the Pearson's correlation between the predicted and observed scores in these tasks. To test the significance of the predictive model with different input feature vectors, we calculate the Fisher z-values of the difference between pairs of correlation values in the same task (see "Methods" for the formula).

Here we choose the fluid intelligence and cognitive flexibility as two representative tests considering their comprehensive involvement of multiple brain system during execution. The control theoretical measurement is more advantageous in predicting the fluid intelligence (Control: $z = 8.12$, $p = 4.44e{-}16$, $n = 865$, Graph: $z = 4.99$, $p = 6.03e{-}7$, Difference: $z = 2.22$, $p < 0.05$, $n = 865$) while the graph measurement leverages more on the cognitive flexibility (Control: $z = 3.63$, $p = 2.84e{-}4$, $n = 865$, Graph: $z = 6.02$, $p = 1.72e{-}9$, $n = 865$, Difference: $z = 1.69$, $p < 0.1$, $n = 865$). The p-values are given by the standard two-tailed tests on the normal distribution. To further examine how each region contributes differently to the predictive model, we define the regional importance z-value as the decrease of model performance when the selected region is removed from the input feature vector. The importance z-value for each of the 8 networks is defined similarly. For the fluid intelligence score, the distribution of importance value (Fig. 4a, b) confirms the involvement of lateral PFC and suggests the attentional network's role in gating the abstract reasoning process[51]. The control theoretical measurements characterize the system's capacity of reaching certain ranges of states with a unit of input energy, analogous to the attentional control view of fluid intelligence where the brain regions in the attentional control network drive the brain to activate the relevant neural systems[52]. For the cognitive flexibility score, in addition to the involvement of attentional network, the graph-theoretical measurement of frontal-parietal network (Fig. 4c, d) is also highly important in capturing the individual difference[53].

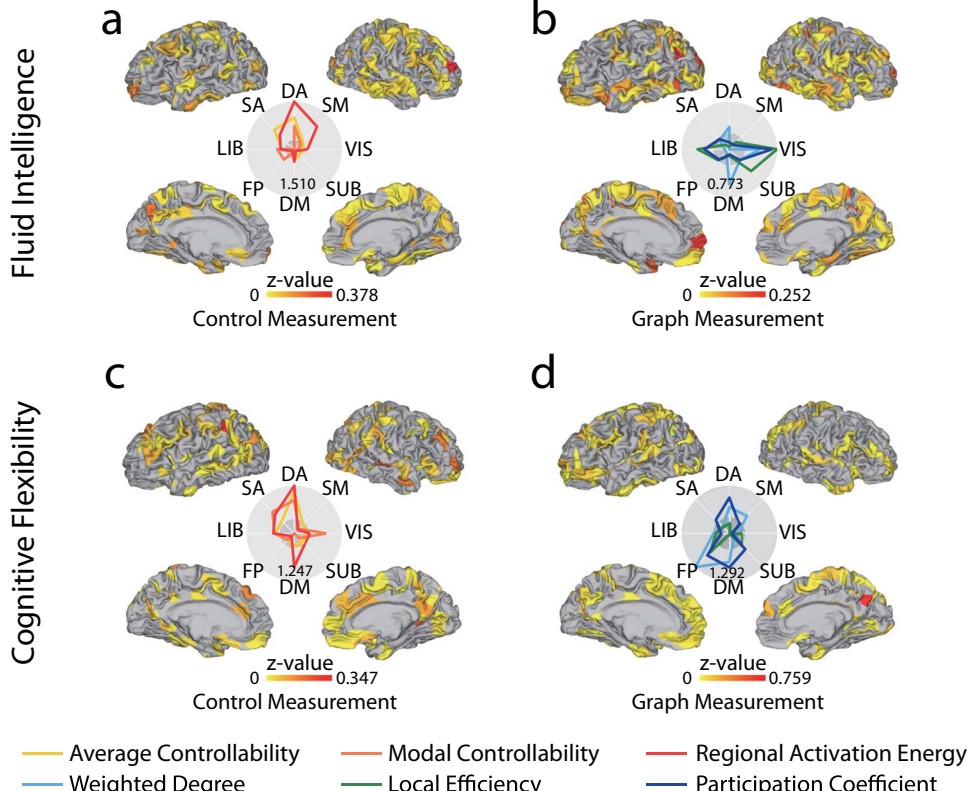

**Fig. 4 Prediction of scores in the behavioral tasks.** The z-value is the Fisher z-value of the correlation between the observed and predicted scores. **a** The control theoretical measurements of the dorsal attention network contribute most in the prediction model. The z-value measures the decline of the model performance when the measurement values on the corresponding area are removed from the input feature vector. **b** The graph-theoretical measurement in the visual system contributes most in the prediction model. **c** The control theoretical measurements of the dorsal attention and default mode networks contribute positively high in the prediction model. **d** The graph-theoretical measurement of the dorsal attention, default mode, and fronto-parietal networks contributes positively high in the prediction model.

The advantage of graph-theoretical measurement over the control theoretical measurement suggests that the cognitive flexibility may be more related to the topological structure of the underline brain network rather than the reachability of multiple states. Given the different performance of these two types of features when predicting different behavioral measures, it suggests that they might be complementary to each other and the combination of them might achieve better prediction accuracy.

Moreover, we examine whether the combination of control and graph-theoretical measurements generally gain additional predictive power than using only the control or graph measurements.

For the combination case, we also use the same size of $6N \times 1$ input vector to control the effect of different input feature vector size on the predicted performance. Especially, we concatenate two $3N \times 1$ feature vectors, one from control theoretical measurements and the other from graph-theoretical measurements. These measurements are calculated on either the static or dynamic functional connectivity matrices. From Fig. 5a, we can see that the combination of control and graph measurements can predict the behavioral scores significantly better than either control measurements ($t = 9.41$, $p = 1.64e{-}13$, $n = 865$) or graph measurements ($t = 4.92$, $p = 3.81e{-}6$, $n = 865$). To further investigate whether the improvement is consistent for the tasks that can be originally predicted by either the graph or graph-theoretical measurement, we examine the distribution of the prediction improvement on a distinct group of tasks. The same improvement of using combined features exhibits on the three subgroups of tasks as well, no matter the task score can be predicted by control measurements (Fig. 5b, Combination vs. Control: $t = 4.07$, $p = 3.29e{-}4$, $n = 865$; Combination vs. Graph:

$t = 4.84$, $p = 5.63e{-}5$, $n = 865$), graph measurements (Fig. 5c, Combination vs. Control: $t = 5.49$, $p = 6.04e{-}6$, $n = 865$; Combination vs. Graph: $t = 1.85$, $p = 3.81e{-}2$, $n = 865$), or their combination (Fig. 5d, Combination vs. Control: $t = 7.79$, $p = 2.31e{-}9$, $n = 865$; Combination vs. Graph: $t = 5.74$, $p = 9.35e{-}7$, $n = 865$). The prediction improvement in Fig. 5b–d is calculated as the z-values of the difference of correlations between the predicted and observed scores. The *p*-values for comparing the *Combination vs. Graph/ Control* are calculated through the one-tailed one-sample *t*-tests as we are testing the significance of improvement rather than difference. In summary, we can conclude that the cross-type feature combination obtains better performance in predicting behavioral scores than that with the pair of features from the same group, no matter control measurements or graph measurements. It suggests that the two measurements capture different aspects of the brain connectome which are complementary to each other so that the combination can generate more powerful biomarkers in characterizing the brain map with respect to different behaviors. The reproducibility results of the predictive model with respect to the kernel choice and data session are included in Notes S1–S4 and Figs. S2–S6.

## Discussion

The brain is a complex system that enables various behaviors through evolving among multiple cognitive states. On the one hand, the brain's state transition can be described with statistical approaches that summarize the intra-region interaction into functional networks. On the other hand, the reconfiguration of the functional networks is strongly associated with the underlying

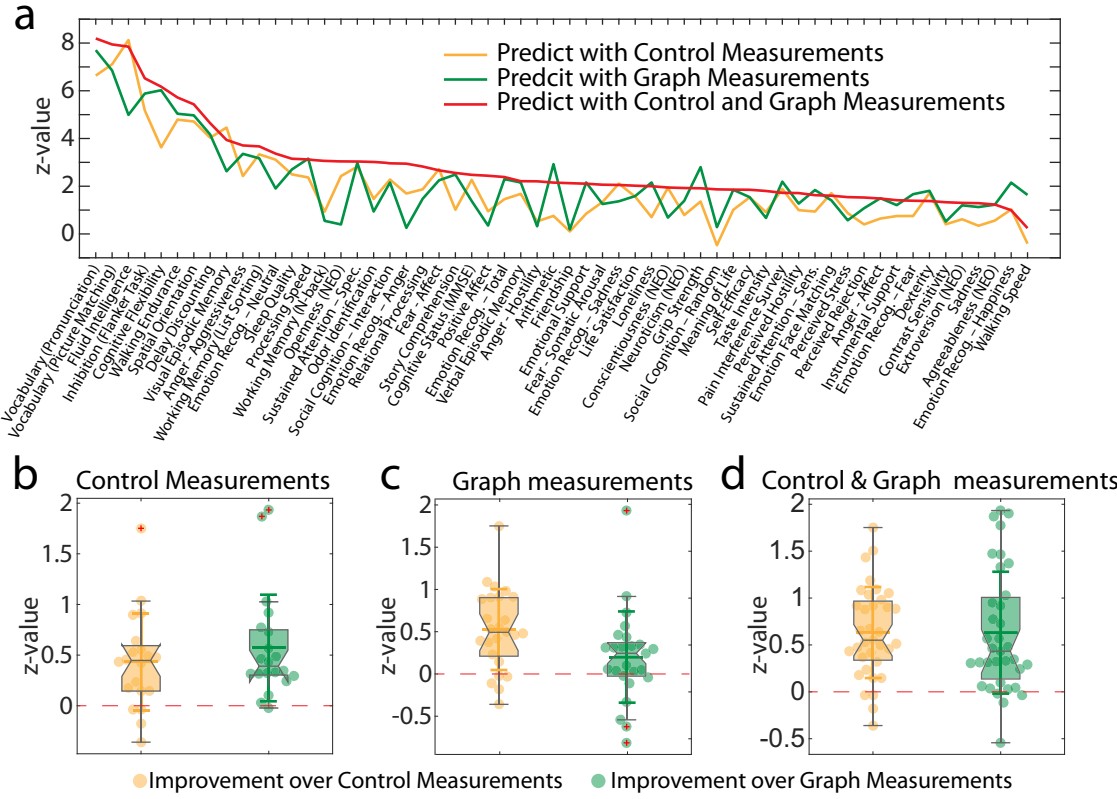

**Fig. 5 Complementary power of graph and control theoretical measurements in predicting behavioral scores.** We demonstrate the gain of predictive power by using a pair of features combining both graph and control theoretical measurements over using features chosen from either graphical or controllability measurements. **a** The improvement is positive for 41 of 58 tasks over the input features selected from either control or graph-theoretical measurements. In (**b**–**d**), we further examine the prediction improvement separately on three groups. We exhibit the distribution of prediction improvement on the task scores **b** that can be significantly predicted by control measurements, **c** that can be significantly predicted by graph measurements, and **d** that can be significantly predicted by the combination of the control and graph measurements. We say the score can be significantly predicted when the correlation of the predicated and observed scores is significant with $p < 0.05$. The prediction improvement in (**b**–**d**) is calculated as the z-values of the difference of correlations between the predicted and observed scores.

transition of cognitive states. Building analyses of dynamics with network constraints shed light on our understanding about both how the network reconfiguration facilitates the state transition and how the efficiency of dynamics are associated with cognitive performance.

The controllability on structural brain networks characterizes the regional ability of alternating large-scale neural circuits based on the assumption that the transition of brain states can be modeled by the structural connectivity[20]. This assumption is supported by the coupling relationship between functional and structural connectome[35,54] and the view that static functional connectivity can be derived through the fluctuation along the structural connectivity[55]. When we consider the control of dynamics on the functional network, the first component is the diffusion that transits under the constraint of the static functional connectivity map. Complementary to static functional connectivity that summarizes the interaction of regions averaged through the whole time course, dynamic functional connectivity describes the temporal causal relationship that is also interpreted as effective connectivity[56]. Thus, in addition to obtaining a model that dismisses the difference over functional states, it is critically significant to model this non-random diffusion and quantize each region's impact of driving the dynamics according to the effective transition pattern. This is why we adopt both the accumulated dynamic functional connectivity matrix from autoregression[38] and the full sequence with the sliding-window approach[6] to

model the time-invariant and variant transitions correspondingly. Considering the brain signal evolution is highly entangled of multiple types of dynamics, we thus propose the current approach of bridging the dynamics and connectivity from different views induced by different types of interaction matrices. This analysis also suggests that brain control model can potentially serve as a fundamental backbone for describing the trajectory of dynamic functional connectivity.

Instead of staying rigid or in a stationary mode, the brain alters its states even in the resting state. This variation brings benefit to the state transition as the time-varying interaction matrix can lead to shorter control trajectories thus lower control energy[49]. This property is also identified in applying control theory to model the suppression of seizures[57]. Indeed, the proposed framework of analyzing the network's efficiency through its energy of state transition provides a quantitative approach to model how the reconfiguration in connectivity is linked to the network's ability in supporting task execution. Previous works have shown that the dynamic connectivity[6] or reconfiguring flexibility[58] can predict the cognitive performance like fluid intelligence[59], n-back memory tasks[60] differently from the static connectivity. Based on what we have built in the current work, the static control can be viewed as a backbone that determines the overall efficiency while the dynamic fluctuation gives how efficient the reconfiguration is executed on the backbone. An intuitive interpretation here would be that higher flexibility or transition rate in brain networks

suggests higher efficiency in utilizing the energy to support the configuration thus better performance in the cognition. This mechanistic interpretation potentially opens a novel way of understanding the brain's functional connectome.

Controllability displays a complementary prediction power to the examined graph-theoretical measurements. Why? On the one hand, in terms of connectome-based models, as shown in previous literature[61,62], the functional connectome encodes the information of individual difference. On the other hand, from the perspective of system control, higher controllability indicates more efficient utility in executing the mode thus the coefficients on the task-related area would contribute positively in the predictive model[30,63]. Although the network biomarkers have been widely used in many situations to understand the brain's connectivity, one necessary yet typically ignored assumption is that the measurements are dependent on certain dynamics on the network. This type of dynamics can be the random walk, greedy diffusion along short path, or other implicit forms. The control paradigm places a linear dynamical system on the brain network allowing each region to impact other region's evolving dynamics based on their coupling strength, which is not covered by previous nodal measurements. This might explain why the two types of measurements can complement each other in the prediction task although the measurements themselves could be highly correlated for their regional distribution[64]. With regard to the control measurements used in the prediction model, we adopt the ones theoretically derived from the connectivity matrix (sFC-Correlation and dFC-Autoregression) rather than the state-dependent ones (dFC-Slidingwindow) in characterizing the state transition to technically equalize the feature vector length in the comparison between control and graph-theoretical measurements. This setup may limit the representativeness of control theoretical features as it suppresses the full-sequence information and it is worth future exploration on how to combine different type of features across multiple modalities in a more comprehensive manner[65]. For now, in the task of predicting the behavioral measurements, the current results suggest that the control metric can provide complementary prediction power to the traditional graph metric, probably from its distinct modeling of the underlying network dynamics.

The controllability of functional brain networks is closely related to that of structural networks. Both frameworks rely on the linear dynamics, which could be limited in capturing the nonlinearity of brain system yet still provides a fair estimation both from the perspective of behavior and control theory[66]. Second, the average controllability displays strong positive correlations with nodal connectivity strength in the structural networks and with effective connectivity strength in the functional networks. This is supported by the results revealed in ref. [67] that the structural connectivity can act as a predictor of the effective connectivity, which also provides the consistency between the current framework and the controllability analysis on structural brain networks[20]. In addition, this positive correlation suggests that although not exactly overlapped, the structural and functional hubs maintain efficient roles in driving the brain to many easily reachable states, providing an explanation of the cognitive association for both structural and functional hubs[68,69].

However, the differences also exist between the two proposed networks. From the modeling perspectives, as the functional control model fits the BOLD time series directly while the structural control model studies the induced dynamics, the two frameworks are generally only applicable to their own modalities. The structural controllability framework implicitly assumes that the signal processes through the physical links and is analogous to the concept of effective connectivity. It provides a mechanical explanation of how the underlined structure supports the

executive function[63] and neural development[24], as well as the evolution of dynamic trajectories associated with the state transition[22]. Due to the limitation in modeling the dynamics from structural connectivity, it is difficult to investigate the role of the control set in driving the whole neural circuit to move across specific states when applying a fixed interaction matrix. The functional control theoretic model implicitly assumes that the random diffusion of brain signal is characterized by the static functional connectivity[70] and the non-diffusive part can be inferred from the time-variant functional connectivity[71]. Thus the proposed control theoretical analyses based on both static and dynamic functional connectivity potentially bridge the gap by analyzing the time series directly rather than inferring from the structure[27]. This is also adopted in a recent work[57] but with effective connectivity. The functional control framework allows future application of intervening the neural circuits via certain nodes for psychiatric medication[72].

Methodologically, it is worth pointing that the current effort still shares certain limitations before. First, by changing the dynamics into a stochastic one, the nonlinear effect still remains to be solved in future. Second, due to the constrains of unpredictable noise in the measurement, the approximation to the observed trajectories is still unsatisfactory. Finally, the controllability measurements are highly related to the effective and functional node strength. Although the linear dynamics could predict the trend as we show in the article, quantitatively, the amount of modeled dynamics is still around the limit point of the linear system thus may not be able to quantify the long-range high-level dependence on connectivity.

In conclusion, modeling the brain network as a controllable dynamic system helps shed light on understanding the brain's transitions between distinct cognitive states. Our findings show that dynamic reconfiguration of functional connectivity permits an optimization of the energy required to maintain the brain's resting state. When compared to standard graph measurements, these energy-based control theoretic measurements also explain complementing variance of individual differences in behavioral scores. These findings pave the way for future research into the relationship between brain's functional controllability diagnostics and clinical variables.

## Methods

**Preprocessing of fMRI data**. We use the HCP 1200-subjects release resting-state functional fMRI data and the behavioral measures of healthy young adults[40]. Informed consent was obtained from all HCP participants. Every subject has four 14.4 min runs (1200 frames) of resting-state functional series, which have a temporal resolution of 0.72 s[42]. Resting-state fMRI data is projected to the fs_LR surface space by the multimodal surface matching method (MSM-ALL[73,74]). Then ICA-FIX[75,76] method is used to clean the imaging data, including the regression of 24 motion-related parameters (6 classical motion parameters, their derivatives, and the squares of these 12 parameters). Frames with FD > 0.2 mm or DVARS > 75 are removed by motion censoring. One frame before and two frames after them are also removed[77,78]. Discard the remaining segments containing less than five frames as well as runs with >50% of censored frames. Regression coefficients are calculated ignoring the censored frames, and linear trends are regressed. Then mean cortical grayordinate signal regression is performed to strengthen the correlation between functional connectivity and behavioral measures[79]. We use a template of 419 regions of interest (ROIs), including 400 cortical areas[43] and 19 subcortical areas defined in Freesurfer[44].

### Construction of the control model

*Functional connectivity.* We discard the first 50 volumes to suppress equilibration effects and normalize each fMRI time series of 419 regions to ensure the mean of the normalized time series is 0 and variance is 1. Then we calculate three different types of function connectivity. The sFC-Correlation matrix $A_f$ is the typically used functional connectivity that is calculated as the Pearson correlation on each pair of areas across full-time points. The dFC-Autoregression is estimated by the first-order autoregression model (AR-model)[11] on full fMRI time series:

$$x(t) = A_\beta \cdot x(t-1) + \epsilon(t), \qquad (1)$$

where $x(t)$ denotes the fMRI activation map (brain states) of all nodes at time $t$, and $\epsilon(t)$ is the residual part. The interaction matrix $A_\beta$ is the model parameter matrix representing the linear dependence of the current state on the previous state. The

dFC-Slidingwindow is the dynamic functional connectivity sequence[7] that uses a rectangular sliding window with a length of 60 TRs and a step of 10 TRs to acquire the subsequences. We calculate the Pearson's correlation on each subsequence to obtain the connectivity matrix within each window.

*Control model.* Following the setup of controllability analysis on the structural brain network[20,22], for a brain system parcellated into $N$ regions, we model the brain state transition through a simplified linear dynamical system given by

$$\dot{x}(t) = A(t) \cdot x(t) + Bu(t), \tag{2}$$

where $x(t)$ is a $N \times 1$ vector that represents the brain states that takes the fMRI activation map at time $t$, $A(t)$ is the $N \times N$ interaction matrix that takes different types of normalized functional connectivity matrix, $B$ is the control input matrix that consists of the column components of the identity matrix to demonstrate the set of control nodes, and $u(t)$ is the input energy over time. For a given type of functional connectivity matrix

$$F(t) = \{F_{ij}(t)\}, \tag{3}$$

we normalize it to a weighted Laplacian matrix to act as the state interaction matrix $A(t)$. Mathematically, we have:

$$A(t) = \frac{-L(t)}{\lambda_{\max}(L(t))}, \tag{4}$$

where $L$ is the Laplacian matrix defined by

$$L_{ij}(t) = \delta_{ij}\sum_k |F_{ik}(t)| - F_{ij}(t), \tag{5}$$

where

$$\delta_{ij} = \begin{cases} 1, & i = j \\ 0, & \text{otherwise} \end{cases}, \tag{6}$$

and $\lambda_{\max}(L(t))$ is the maximum absolute eigenvalue of the Laplacian matrix. We use the absolute value to stabilize the system dynamics. In the current work, the interaction matrix $A(t)$ is time-invariant when we set it as sFC-Correlation or dFC-Autoregression and time variant when we set it as dFC-Slidingwindow.

*Control with static interaction matrix.* When the interaction matrix $A(t)$ is time-invariant, we denote it as $A(t) \equiv A$. Then for a given pair of initial state $x_0$ and target state $x_f$, the optimal control energy driving this transition with $\tau^s$ time ($s$ for static) can be calculated by

$$E_s\left(x_0, x_f\right) = \frac{1}{2}d^T W^{-1} d, \tag{7}$$

where

$$d = x_f - e^{A\tau^s} x_0 \tag{8}$$

is the difference between the desired final state $x_f$ and the hypothetical final state that the system will reach from $x_0$ in the absence of control, and $W$ is the controllability Gramian matrix[22]. The regional activation energy for node $i$ with all nodes as the control set is then defined as

$$\varepsilon_i = \frac{1}{2}d_i^T W^{-1} d_i, \tag{9}$$

where $d_i = (0, \ldots, 1, \ldots 0)^T$ is the $i$-th column from an identity matrix denoting the activation on the $i$-th region[64]. We further use two different controllability measurements to quantify each node's control capability of the linear system, namely average controllability and modal controllability. Average controllability equals the average input energy from the given control nodes to drive the state transition to all possible target states. Suppose that we control the whole system from region $i$, the control corresponding control matrix $B_i = diag(0, \ldots, 1, \ldots, 0)$ with the $i$-th diagonal term being 1 and others being 0. The Average Controllability of region $i$ is then calculated as

$$\psi_i = \text{Trace}(W_i), \tag{10}$$

where $W_i$ is the controllability Gramian matrix with respect to $B_i$ in the form of

$$W_i = \int_0^\infty e^{At} B_i B_i^T e^{A^T t} dt, \tag{11}$$

and $e^{At}, e^{A^T t}$ are the matrix exponential. The Gramian matrix $W_i$ can be solved through the Lyapunov equation

$$A W_i + W_i A^T + B_i B_i^T = 0. \tag{12}$$

Modal controllability of region $i$ refers to its ability to control each evolutionary mode of a dynamical network and is given by

$$\phi_i = \sum_{j=1}^N \left(1 - \lambda_j^2(A)\right) v_{ij}^2, \tag{13}$$

where $\lambda_j(A)$ is the $j$-th eigenvalue of system matrix $A$, and $V = [v_{ij}]$ represents the eigenvector matrix of $A$. More detailed derivation of the formulas can be found in the colloquium paper on control theory[80].

*Control with dynamic interaction matrix.* Following the definition of temporal networks[49,81], suppose we have $M$ snapshots and the duration of the snapshots $m$ is $\tau_m^d$ ($d$ for dynamic), we acquire dynamic functional connectivity matrices $A_m$ using the sliding-window method in this part and normalize it as shown in Eq. (4) for the interaction matrix. The snapshot-based time-variant interaction matrix is then given as

$$A(t) = A_k, \tag{14}$$

for $\sum_{m=1}^{k-1}\tau_m^d < t \le \sum_{m=1}^k \tau_m^d, k = 1, \ldots, M$. The minimum dynamic control energy can then be calculated by

$$E_d\left(x_0, x_f\right) = \frac{1}{2}d^T W_{\text{dyn}}^{-1} d, \tag{15}$$

where

$$d = x_f - e^{A_M \tau_M^d} e^{A_{M-1} \tau_{M-1}^d} \cdots e^{A_1 \tau_1^d} x_0 \tag{16}$$

is also the difference between the desired final state $x_f$ and the hypothetical final state that the system reaches from $x_0$ without energy input. The temporal Gramian $W_{\text{dyn}}$ is defined as $SWS^T$, where

$$S = \left(\prod_{l=M}^2 e^{A_l \tau_l^d}, \ldots, \prod_{l=M}^{m+1} e^{A_l \tau_l^d}, \ldots, I\right) \tag{17}$$

and

$$W = diag\left(W_1, W_2, \ldots, W_M\right) \tag{18}$$

is a block diagonal matrix composed of each snapshot's controllability Gramian matrix.

In the experiments for Fig. 3, we randomly select two time points on the normalized fMRI time series and construct the static and dynamic control systems with the time series between the selected two time points. We set the initial state $x_0$ and the final state $x_f$ as the average value of the middle 10 time points in the initial and final snapshot. In the dynamic control system, we consider the dynamic duration $\tau^d$ of the snapshots at both ends ($\tau_1^d = \tau_M^d = 30$), the period of the snapshots in the middle $\tau_i^d = 60$. In contrast, the duration of static control $\tau^s$ is the sum of the time of all $M$ snapshots. We repeat 12 times on each subject to compare the minimum static control energy and the minimum dynamic control energy. And for the dynamic control system, we randomly shuffle the order of snapshots 20 times to compare dynamic control energy under different orders. To obtain the relationship between minimum control energy and duration, we randomly set the control duration $\tau^s$ from 1 to 1000 as the static system duration and scale each snapshot's duration $\tau^d = \tau^s/M$ correspondingly. We use $E_s$, $E_d$, and $E_r$ for the control energy with respect to static, dynamic, and random dynamic functional connectivity matrices, respectively. In terms of the average controllability and modal controllability for control with dynamic interaction matrix, the two measurements are first calculated within each snapshot with interaction matrix $A_m$ and then averaged over the $M$ windows as the controllability measurements for the whole time-variant linear control system.

**Prediction and feature importance**. For the static and dynamic functional connectivity matrices, we calculate each regions' weighted degree, local efficiency, and participation coefficient as the graph-theoretical features. On the other side, we use average controllability, modal controllability, and regional activation energy as the controllability features in the form of the control feature vector $(\varepsilon_1, \ldots, \varepsilon_N, \psi_1, \ldots, \psi_N, \phi_1, \ldots, \phi_N)^T$ of the size $3N \times 1$ with $N$ as the number of regions. We considered the autoregression and static models for two reasons. First, both autoregression and sliding-window methods are to model the dynamic functional connectivity. Second, sliding-window method leads to multiple windows where the graph structure may be very noisy and makes the calculation of traditional graph measurements unreliable. For the sake of better alignment in terms of the number of features and fair comparison between control and graph-theoretical measurements, we only adopt the static and autoregression ones here.

Then we build a prediction model using the kernel ridge regression method[50] with the cosine kernel and the 10-fold cross-validation to predict the 58 behavioral measures. Here we use the out-of-sample prediction with equal number of features in the compared models to eliminate the need to control for the number of parameters. Under this setup, the model predicts on the part of data not accessible during the training step so that complex model automatically gets penalized for overfitting.

In terms of the null model, we randomly shuffle the correspondence between the measurement values and the subject and perform the same analyses through the prediction model described above. To evaluate the regional importance in the prediction model, we iteratively remove the input feature of each region and calculate the change of correlation in the form of Fisher z-values.

**Statistics and reproducibility**. The $t$- and $p$-values in Fig. 3a, e are obtained by the one-sample $t$-test[82] that tests versus the null hypothesis that the energy difference is

mean zero. The r- and p-values in Fig. 3f–h are the standard Pearson's correlation and the associated *p*-values. For the results in Fig. 4, the z-values for the difference in two cases are calculated in the follow steps. First, we transform the correlations using Fisher z transformation:

$$z_i = \frac{1}{2}\ln\frac{1+r_i}{1-r_i} \text{ for } i = 1, 2. \quad (19)$$

for $i = 1, 2$. Next, the test statistics of the difference of two correlations are calculated by

$$z_d = \frac{z_1 - z_2}{\sqrt{\frac{1}{N_1-3}+\frac{1}{N_2-3}}}, \quad (20)$$

where $N_1 = N_2$ is the number of subjects[83,84]. For the results in Fig. 5, the z-values of the predication difference are calculated in the same way as in Fig. 4. The *t*- and *p*-values in Fig. 5b–d are calculated with the one-tailed one-sample *t*-tests. We choose the one-tailed set here as we are interested in whether the combination of control and graph measurements outperforms each separately rather than whether they are different from each other.

Regarding repeatability, given that HCP encompasses four sessions for each subject's resting scan, we use the first session to describe the results and the other sessions to analyze the reproducibility of the primary findings. Figure S2 illustrates the similarity of control and graph measurements across four sessions. Figure S4 and Fig. S5 illustrates the similarity of predicted scores across four sessions with cosine or linear kernels, respectively.

**Our workflow**. After preprocessing, we obtain the fMRI time series. The analytic pipelines are as follows:

*Connectivity*. The static functional connectivity is calculated using Pearson's correlation and the dFC-Autoregression is calculated using the 1st-order autoregression model. The dFC-Slidingwindow is a dynamic functional connectivity sequence that acquires subsequences using a rectangular sliding window with a length of 60 TRs and a step of 10 TRs.

*Control model*. The Laplacian matrix is computed using Eq. (4). Then following Eq. (2), we construct the control model, where $x$ denotes the brain state, and $A$ is the Laplacian matrix derived from the three types of connectivity matrices. The input matrix $B$ is constructed using the identity matrix's column vectors, and $u$ denotes the system input.

*Controllability measurements*. We calculate the controllability measurements, including the average controllability (Eq. 10) and the modal controllability (Eq. 13) from the three types of functional connectivity. Then we compute the mean average controllability and modal controllability across all subjects and plot their spatial distribution (Fig. S1). The regional measurements for dFC-sliding-window sequence are averaged over all sub-windows. Following that, we calculate the average functional controllability of each subsystem by averaging all the areas inside the subsystem and display the histogram and its variance (Fig. 2).

*Control energy*. Given an initial state $x_0$ and a final state $x_f$, we compute the static control energy consumption $E_s$ by Eq. (7) and the dynamic energy using the approach described in the section "Control with dynamic interaction matrix".

*Static control and dynamic control*. We extract time snapshots from the BOLD time series with a window size of 60 and calculate both static (for the whole time series) and dynamic function connectivity (for each snapshot). Then, using the approach described in step 4, we compute the static and dynamic control energies for various hypothetical time intervals $\tau_s$. We repeat 12 times for each subject with different $\tau_s$. In Fig. 3, we set the initial and final states as the average value of the middle 10 BOLD time points in the first and last snapshot. In Fig. S7, the initial state is determined directly by the BOLD value of the first snapshot's start point, and the final state is determined directly by the BOLD value of the last snapshot's endpoint.

*Disrupting the order of snapshot*. We randomly sort the snapshots 20 times and then calculate the dynamic control energy associated with each order. We compare the increment ratio after disrupting the snapshots and the dynamic control energy associated with the original order.

*Prediction of scores in the behavioral tasks*. We employ three standard graph metrics in this section: the weighted degree, the local efficiency, and the participate coefficient. We compute the average controllability, modal controllability, and regional activation energy for the control measurements. Each element $e_i$ of the regional activation energy is the static control energy that from the initial state $(0, 0, \dots, 0)^T$ to the final state $(0, 0 \dots, 1, \dots 0)^T$ (the *i*-th element is 1) when the entire brain acts as the driving nodes. We compute each measurement using static functional connectivity and dFC-Autoregression, and then combine the measurements with the same name in both cases to obtain the prediction feature vector. We can predict behavior tasks using kernel ridge regression and tenfold cross-

validation on the feature vectors. As the prediction performance index, we utilize the Fisher z-value (statistical tests section in the main text method) derived from the correlation between the expected result and the true value. Then the drop in prediction performance caused by the deletion of subsystem sections is utilized to determine the subsystem's contribution to the whole prediction.

*Complementary prediction power of graph and control measurements*. We combine the control and graph measurements in a new feature vector and predict the behavioral task scores (the same with step 7). Following this, we compare the prediction performance of the combined feature vector with that of the feature vector constructed from control or graph measurements (Fig. 5).

**Reporting summary**. Further information on research design is available in the Nature Research Reporting Summary linked to this article.

## Data availability
The Human Connectome Project Data is publicly available online at http://www.humanconnectomeproject.org/data. Source data for all figures are provided in Supplementary Data 1–4.

## Code availability
The code is available at https://github.com/Gus-Lab/functional_controllability.

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

## Acknowledgements

S.G. and S.K.D. are supported by National Natural Science Foundation of China (General Program, 60876032). B.T.T.Y. is funded by the Singapore National Research Foundation (NRF) Fellowship (Class of 2017), the NUS Yong Loo Lin School of Medicine (NUHSRO/2020/124/TMR/LOA), the Singapore National Medical Research Council (NMRC) LCG (OFLCG19May-0035), and the Singapore National Supercomputing Centre. Any opinions,

findings and conclusions or recommendations expressed in this material are those of the authors and do not reflect the views of the Singapore NRF or the Singapore NMRC. J.L. is supported by the Deutsche Forschungsgemeinschaft (EI 816/4-1), the National Institute of Mental Health (R01-MH074457), the Helmholtz Portfolio Theme Supercomputing and Modeling for the Human Brain, the European Union's Horizon 2020 Research and Innovation Program under Grant Agreements No. 945539 (HBP SGA3) and 826421 (VirtualBrainCloud).

## Author contributions

S.G. designed the research; S.K.D. and J.W.L. processed data and performed research; S.G., S.K.D., J.W.L., and B.T.T.Y. contributed new analytic tools and wrote the paper.

## Competing interests

The authors declare no competing interests.
