## [Peer Review File · Communications Biology]

Reviewers' comments:

Reviewer #1 (Remarks to the Author):

This article proposes modeling the brain resting state activity using the dynamic controllability of the regional networks. In the first part, the authors compared both the average and modal controllability when the control is induced using static, autoregression, or sliding window connectivity. In the second part, they studied how control and graph-based metrics can predict behavioral measurements.

As a general comment, this project takes roots in 3 articles of the authors (ref11/ref16/ref32) with the idea to compare different modeling. It is indeed exciting research, but they failed to conclude each model's relative merit. We end up not understanding what should be used and why. For predicting the behavioral measurement, the relative merit of the control metric over the graph metric was not clearly stated.

The abstract is lacking any results, and some parts are cryptic or not based on results. What is the meaning of "benefit neural processing" or "facilitates the cognition" ?; Both are nice expressions, but their meaning should be explained.

In the introduction, the white matter network (paragraphs 94-109) will be the basis of the modeling; however, from what I understand, contrary to their previous work (16), the current modeling was solely based on functional connectivity. It is a weakness of the work, as the results seem different from what they previously published (figure 2 versus figure 2 from ref16).

As a general comment about statistics, it is not easy to understand the meaning of p-value at 10 to the power -16263 (for example, line 291); it does not mean anything and is beyond the precision that can be computed. The N is probably huge and arbitrary set; in this vein, it is complicated to be convinced by the regression of fig3.h. I am expecting the statistic to be based on the variability across the subjects. In the same way, any comparison between models should consider the number of parameters (For part 2), criteria such as Akaike (and there are others) had been designed for such purpose. There is also some implicit reference to statistics that are just observation-based and not very clear, see the comment on the significance on line 282 for fig3.d.

The authors should state more clearly what are the average and modal controllability and does it says on how the brain is working. This is essential as the best model for the modal controllability (autoregression) is also the worst for average controllability. The figure showing the regional metrics (fig2a and b) is difficult to interpret; the color scale is not the same. What is the meaning of the two regions that exhibit a "hot spot" in the average controllability by sliding window? The results are interpreted only in terms of the eight networks; thus, I fail to see the interest to keep parts a and b of figure 2. The difference between the model should be stated using statistical analysis and across subjects (not across the number of paths). For example, I wonder if modal connectivity is different between auto-regression and static.

In the behavior prediction analysis, The author should justify why they used both autoregression and static but not sliding-window, which they show to be the best modeling for the average controllability in the first part. They considered that the graph measurement leverage more on the cognitive flexibility with a $p < 0.1$ (figure 4d and line 348); however, a p at 0.1 is a nonsignificant result. The 3D rendering seems to be just for illustration purposes as the results are only interpreted for the resting-state networks. Improvement of one model over the other should be acknowledged by the proper statistical procedure considering the number of parameters in the model (see above).

Minor :

The sentence line 295-297 ("This indicates that ...") is difficult to understand.

The sentence on line 234 should state that the authors commonly use the eight brain networks.

The figure legends should be shortened.

Why does the author choose the notation "6Num_of_Regions x 1 "

typo :

line 299, figure 5 "predict", line 345 reference should be to figure 4a and not 3a

Reviewer #2 (Remarks to the Author):

In the manuscript titled "Energetic advantage in control dynamics explains dynamic reconfiguration traits during the resting state" the authors present an extension of work applying network control theory to functional-MRI by exploring the impact of replacing typically used static measures of structural connectivity as the interaction matrix in this framework with static and dynamic formulations of functional connectivity (FC). The authors show average controllability of the dynamics is higher when using sliding-window dynamic FC and modal controllability is higher when using autoregression dynamic FC. The default mode and frontal-parietal networks show a large influence on average controllability in the dynamic case and conversely the sub-cortex demonstrated large influence on modal controllability. Combining these measures with graph theory metrics is then shown to improve prediction of behavioural scores.

Overall the paper is interesting and timely, both in methodology and results. After sufficient response and changes to my comments below, I recommend the paper for publication.

Key points:

1. In general the mathematics are not clearly explained as they have been applied. This forms a key part of interpreting the validity of the results in the context of the neuroscience. The method needs to be reproducible and as it stands I do not think this is the case. The mathematics is straightforward and thus a more text throughout the manuscript, but in particular the method section, describing its application to the timeseries data is needed.

2. How do these functional connectivity results contrast those where the interaction matrix is a measure of structural connectivity?

3. Line 252: "Energy Efficiency Explains the Dynamic Reconfiguration of Functional Connectivity." The word 'Explained' is too strong here

4. Fig 3: This caption is insufficient. Explain what is plotted. Discussion of results is the purpose of main text.

5. Line 256: What is randomly shuffled?

6. Line 272: *significantly

7. Line 586: Do you actually evaluate this integral over $t=0$ -inf or do you impose a time horizon? Please explain exactly how you've implemented the mathematics.

8. Line 606: "We set the initial state x_0 and the final state x_f as the average value of the middle 10 time points in the initial and final snapshot" What does this mean?

9. Line 606: The idea behind this paragraph is good, i.e., walk through the mathematics as you have applied it, but the descriptions given here are muddled and full of jargon not defined. A supplementary figure detailing the methods application to the timeseries would be helpful.

10. Fig. 4d *participation coefficient

Reviewer #3 (Remarks to the Author):

In this work, Deng and colleagues give an analysis tour-de-force to improve our current understanding of brain networks and how they are related to behavior. This is a great example on

how to use large-scale, open-source, neuroimaging databases (in this case HCP, resting-state fMRI) to propose both methodological advances that have testable theoretical implications. I have a few concerns, though, that preclude the acceptance of this paper as it is:

Major

Results

-For the results provided for Fig 1, the data is described as "the average controllability of the dynamics induced from dFCSlidingwindow is higher than that of the sFC-Correlation and dFC-Autoregression (Fig. 2c) except for the subcortical areas" in the results and "In general, the average controllability of the brain system parameterized by dFSlidingwindow is higher than the systems parameterized by sFC-Correlation" in the Figure, but no statistics are provided. If the authors want to keep this analysis as it is, then a better explanation of why no statistical comparison between methods is warranted here.

-The authors say "Here we choose the fluid intelligence and cognitive flexibility as two representative tests considering their comprehensive involvement of multiple brain system during execution". As seen in Figure 5a, there are many other behavioral scores that could have been selected as they also involve multiple brain systems during execution (sustained attention, working memory, as an example). The authors need to justify better why cog flexibility and fluid intelligence were chosen or why performance is worse for other behavioral scores that should also depend on multiple brain systems.

Methods

-“We choose the one-tailed set here as we are interested in whether the combination of control and graph measurements outperforms each separately rather than whether they are different from each other”. This needs a better justification, taking into account that two-tailed t-tests are used in other analysis.

Minor

-Fig3. The pvalues in the figure are difficult to see

-“This might explain why the two types of measurements can complements each other” should be complement.

**Control Theory Illustrates the Energy Efficiency in the Dynamic
Reconfiguration of Functional Connectivity**

Shikuang Deng, Jingwei Li, B.T. Yeo, Shi Gu

We thank all the reviewers for their insightful comments and highly constructive suggestions to help improve the quality of our work. We have carefully addressed all the comments and revised the manuscript accordingly. We believe that the manuscript has been improved significantly according to the suggestions, and we are grateful for the reviewers' helpful comments. In the following context, we will address the comments from each reviewer one by one.

Reviewer 1

Summary. "This article proposes modeling the brain resting state activity using the dynamic controllability of the regional networks. In the first part, the authors compared both the average and modal controllability when the control is induced using static, autoregression, or sliding window connectivity. In the second part, they studied how control and graph-based metrics can predict behavioral measurements."

Response: We thank the reviewer for the highly constructive feedback. In the revision process, we address the comments piece by piece and respond to each comment in detail below.

Comment 1. "As a general comment, this project takes roots in 3 articles of the authors (ref11/ref16/ref32) with the idea to compare different modeling. It is indeed exciting research, but they failed to conclude each model's relative merit. We end up not understanding what should be used and why. For predicting the behavioral measurement, the relative merit of the control metric over the graph metric was not clearly stated."

Response: We appreciate the comment and are happy to clarify our major contribution. In this paper, our main purpose is to establish a paradigm of applying brain control theoretic analyses on the functional MRI data. In order to demonstrate that the proposed control analyses bring unique methodological values beyond the traditional network approaches, we performed the analyses on explaining brain state transition and predicting behavioral measurements. Considering the scope of the current work, we only showed the usefulness and the complementary power of control theoretic measurements over other approaches. We agree with the reviewer that a detailed discussion on "what and why" would be very important and exciting but that may take additional effort of a separate paper like ref11. Based on the current content, our conclusion is that *Controllability displays prediction power complementary to the examined graph theoretical measurements*. This was discussed in lines 457-478 in the

previous version.

To explicitly state the relative merit of control theoretic analyses for modeling dynamic functional connectivity, we add a conclusive sentence on lines 487-489 in the revised manuscript. For the ease of review, we paste it below.

For now, in the task of predicting behavioral measurements, the current results suggest that the controllability metric can provide prediction power complementary to the traditional graph metrics, potentially due to its distinct modeling of the underlying network dynamics.

Comment 2. *“The abstract is lacking any results, and some parts are cryptic or not based on results. What is the meaning of “benefit neural processing” or “facilitates the cognition” ?; Both are nice expressions, but their meaning should be explained.”*

Response: We thank the reviewer for pointing out the content that may cause confusion in the abstract. In the revised manuscript, we delete the sentence *“Understanding how this specialized reconfiguration of the brain’s functional network mechanistically benefits neural signal processing and facilitates cognition from the dynamical system perspective lies on the frontier of network neuroscience.”* from the abstract and add an explanation on lines 79-82. For the ease of review, we paste the updated paragraph below.

While these works provide insightful description of the functional connectome separately from the statistical and mechanistic views, the isolated studies on statistically defined connectivity and the rich dynamics of functional neuroimaging leave it vacant to mechanistically explain the principle of dynamic reconfiguration^{8,9}. Specially, it is not well understood how the specialized reconfiguration of the brain’s functional network leads to high network efficiency that mechanistically benefits neural signal processing. Furthermore, considering that the brain activity involves both the metabolic and signal-spreading procedure, this integration of the statistical and mechanistic views may generate novel network-based biomarkers that improve the predictive power for individual differences in human behaviors¹⁰⁻¹². Thus, it is of critical significance to build a dynamical model that can illustrate how the reconfiguration of the brain’s functional networks can facilitate the state transition in the brain. Furthermore, the proposed dynamic model should be able to derive effective biomarkers with complementary power to traditional graph theoretical measurements when predicting human behavioral scores¹³.

In terms of the comment *“The abstract is lacking any results, and some parts are cryptic or not based on results.”*, we assume the reviewer thought we did not add the quantitative results in the abstract. We really appreciate this suggestion and understand the reviewer’s concern. But our consideration here is that the major contribution of the current paper is to establish the control analyses on functional

brain networks rather than a purely technical improvement on the prediction accuracy. Thus, we summarize our contribution more from the methodological aspects for a broader range of audience.

Comment 3. *“In the introduction, the white matter network (paragraphs 94-109) will be the basis of the modeling; however, from what I understand, contrary to their previous work (16), the current modeling was solely based on functional connectivity. It is a weakness of the work, as the results seem different from what they previously published (figure 2 versus figure 2 from ref16).”*

Response: We thank the reviewer for this comment and apologize for the confusion in the context. We want to clarify that the current work is based on connectivity while other existing works on network control theory are built upon the white matter network. Our major motivation of developing the current framework on functional data is to address the limitation of the previous work on the dynamics of functional connectivity. In addition, although there have been a large amount of work showing the coupling relationship between structural and functional connectivity, it remains unclear how to or whether it is possible to determine functional connectivity from structural connectivity. Thus, it is necessary to provide a control theoretical analysis on the functional networks directly besides the derivative model from structural brain networks. Since the controllability measurements are calculated based on different types of networks, the “inconsistency” is not a weakness but an expected result given differences between the structural and functional connectome.

Finally, we agree with the reviewer that it would be great to incorporate structural connectivity into the model to account for structure-function coupling. We will investigate this setup in future studies.

To make it clear why we build the control model on functional connectivity rather than adopting the one with structural connectivity, we re-organize the contexts on lines 90-104 so that the value of control theory and its shortcoming with structural assumption are more explicitly stated. For the ease of review, we paste the contexts below.

*Network control theory is a particularly promising mathematical framework to model the structure-induced brain state reconfiguration ^{14,15}. Originally developed in the physics and engineering literature ¹⁶, the approach is flexible to applications across scales and species, including cellular models ¹⁷, *C. elegans* ¹⁸, fly ¹⁹, mouse ¹⁹, macaque ²⁰, and human ²⁰, and has been extended to study cognitive function²¹⁻²³, development ²⁴, heritability ²⁵, disease ^{26,27}, and the effects of stimulation ²⁸⁻³². Such broadening utility of network control theory in neuroscience is built on the empirical estimate of the network’s structural architecture and the modeling of the dynamics that such an architecture can support ^{19,20}. On the one hand, the dependence on structural connectome brings convenience in characterizing the evolving whole-brain states through analytical tools. On the other hand, current works utilizing notions of network control are to some degree limited by the*

assumption that all effective relations between regions are time-invariant and encapsulated in the underlying white matter network architecture. This assumption leaves the approach agnostic to distinct ways in which the connection diagram can be utilized for inter-regional communication³³, both to support diverse states in health²¹ and disease³⁴.

Comment 4. “As a general comment about statistics, it is not easy to understand the meaning of p -value at 10 to the power -16263 (for example, line 291); it does not mean anything and is beyond the precision that can be computed. The N is probably huge and arbitrary set; in this vein, it is complicated to be convinced by the regression of fig3.h. I am expecting the statistic to be based on the variability across the subjects. In the same way, any comparison between models should consider the number of parameters (For part 2), criteria such as Akaike (and there are others) had been designed for such purpose. There is also some implicit reference to statistics that are just observation-based and not very clear, see the comment on the significance on line 282 for fig3.d.”

Response: We thank the reviewer for the suggestions and would love to clarify our setup of statistics here.

In terms of the results in Fig. 3, considering that the energy associated with trajectory is affected by not only its initial and target states but also individual differences, we chose to sample multiple sliding windows and report over all sampled pairs across the whole population, which led to the large N and overly significant p -values. Indeed, we would more like to interpret the results in Fig.3 b-d to be driven by some physical principles rather than statistical variance across different subjects. To relieve the reviewer’s concern, we divided the τ^s into three intervals and report the comparison significance within in each interval as well as the whole range. See below for the detailed values please. We also include this in the supplement as well.

	$\tau^s < 10$		$10 \leq \tau^s < 100$		$\tau^s \geq 100$		$\tau^s > 0$	
	T value	p-value	T value	p-value	T value	p-value	T value	p-value
DM	97.2	2.14e-188	66.66	1.90e-492	89.5	6.06e-979	99.53	7.61e-1422
DM,DA,SA,FP	54.44	3.02e-36	121.01	2.57e-827	260.24	2.29e-4195	289.85	1.48e-4964
ALL	29.02	3.70e-08	147.41	1.80e-638	346.58	2.05e-3513	369.92	2.91e-3837

Table. S3. Student’s t -test results of the energy efficiency comparison within different intervals of control time (Fig 3.b-d).

In terms of Fig.3 f-h, we agree with the reviewer that it would be informative to provide the results over all subjects. Thus, we added additional experiments where we averaged the quantities in Fig.3 b-d across states first and only regressed across

subjects. The conclusion remained the same. We include this result as Fig. S8 in the supplement.

DM: Default Mode, DA: Dorsal Attention, SA: Saliency, FP: Fronto-parietal, CN: Combined Network, All: Whole Network

Fig. S8. *The energy efficiency of functional connectivity order across subjects. The static results across subjects remain unchanged from Fig. 3 in the main text. The relative increase of log-energy is negatively correlated to the log-energy of dynamic control.*

In terms of the model comparison, we totally agree with reviewer that any comparison between models should consider the number of parameters. Indeed, this was also a critical factor we consider when we built our analyses of predicting the cognitive task scores. We did not use any of the statistical penalty for model complexity here since all of them are based on certain statistical assumption of parameter and sample distribution, which may not be satisfied in our application. In the current work, we considered two steps to take care of these issues instead. First, we ensure that the parameters in the two prediction models are same, i.e. we set the control and graph measurements to be of the same length. Through this setup, the model complexity is fair for the two types of measurements. Second, we report the values through K-fold cross-validation rather than making penalty solely on the training set. Under this setup, the model predicts on the part of data not accessible during the training step so that complex model automatically gets penalized for overfitting. Out of these two reasons, we provided a more realistic data-driven solution for the model comparison in the current project thus did not include the discussion of using criteria such as Akaike.

We also add the clarification of this point to the section of *Prediction and Feature Importance*. For the ease of review, we paste it below.

Prediction and Feature Importance

...

Then we build a prediction model using the kernel ridge regression method⁵¹ with the linear kernel and the 10-fold cross-validation to predict the 58 behavioral measures. Here we use the out-of-sample prediction with equal number of features in the compared models to eliminate the need to control for the number of parameters. Under this setup, the model predicts on the part of

data not accessible during the training step so that complex model automatically gets penalized for overfitting.

...

Comment 5. *“The authors should state more clearly what are the average and modal controllability and does it says on how the brain is working. This is essential as the best model for the modal controllability (autoregression) is also the worst for average controllability. The figure showing the regional metrics (fig2a and b) is difficult to interpret; the color scale is not the same. What is the meaning of the two regions that exhibit a "hot spot" in the average controllability by sliding window? The results are interpreted only in terms of the eight networks; thus, I fail to see the interest to keep parts a and b of figure 2. The difference between the model should be stated using statistical analysis and across subjects (not across the number of paths). For example, I wonder if modal connectivity is different between auto-regression and static.”*

Response: We thank the reviewer for pointing out this confusion in Fig. 2. According to the reviewer’s suggestion, we changed the Fig.2 to the boxplots with error bar and remove the brain surface plot. The corresponding statistics are included in the supplement Table S1 and Table S2. In addition, we also add statements about average and modal controllability in lines 192-207 as suggested. For the ease of review, we paste the content below.

Revised context:

*Based on the constructed dynamics as well as the associated statistics, we obtain the distributions of average and modal controllability that quantify **the volume of states the system can arrive and distance the system can reach given a unit of input energy** respectively. The 419 regions are grouped into 8 brain networks [43, 50]. And the controllability when setting each region as the controller is averaged within each network.*

Although it remains unknown how the brain intrinsically drives the transition across different states, we can utilize these two measurements to infer each region’s preference in control strategy following the basic rule that the brain must cost energy that depends on connectome and states for the dynamical reconfiguration. If the average controllability in a region is higher, it suggests that energy input on the region would be more efficient if the goal is to cover as more states as possible. Thus intuitively, it quantifies the regional ability in moving the whole brain to many easily reachable states. Respectively, the modal controllability gives a description of the regional ability in moving the whole brain to difficult-to-reach states respectively ¹⁶.

Revised Fig. 2:

Fig. 2. Spatial Distribution of Functional Controllability. We show the distribution of mean **(a)** average and **(b)** modal controllability across different functional brain networks. The mean average control values (Mean ac-Value) and mean modal control values (Mean mc-Value) are calculated across all subjects. The system mean is calculated across the within-system regions for all subjects. We adopt the eight brain networks from Yeo's partition to compare the difference in controllability for different networks ^{44,50}

Additional Results in the supplement:

	Static FC VS. dFC-Autoregression		Static FC VS. dFC-Slidingwindow		dFC-Autoregression VS. dFC-Slidingwindow	
	T value	p-value	T value	p-value	T value	p-value
	Visual	34.66	5.85e-230	-38.90	2.64e-287	-75.53
Somatomotor	53.37	3.41e-326	-13.02	4.93e-62	-83.00	1.64e-402
Dors_Attn	59.38	3.27e-411	-45.68	4.01e-340	-97.43	2.74e-474
Saliency	105.97	5.95e-573	-8.52	3.75e-23	-96.10	4.11e-478
Limbic	94.90	9.42e-517	-5.34	5.36e-10	-82.89	3.54e-423
Fronto_Parietal	44.42	1.46e-300	-106.34	6.25e-615	-143.28	7.96e-633
Default	72.24	1.22e-496	-64.34	2.17e-429	-115.15	3.86e-537
Sub_cor	96.47	2.24e-504	48.02	2.85e-411	-43.43	1.86e-217

Table. S1. Student's *t*-test results of the system-wise average controllability for three cases (Fig 2. a).

	Static FC VS. dFC-Autoregression		Static FC VS. dFC-Slidingwindow		dFC-Autoregression VS. dFC-Slidingwindow	
	T value	p-value	T value	p-value	T value	p-value
	Visual	-30.79	2.42e-225	32.54	4.26e-294	195.78
Somatomotor	-61.66	7.92e-438	3.83	2.58e-7	194.60	3.92e-760
Dors_Attn	-59.62	4.54e-492	38.27	1.47e-353	292.99	2.29e-1051
Saliency	-121.96	4.04e-744	-10.69	2.13e-41	296.54	1.17e-1138
Limbic	-115.88	5.26e-728	-17.33	5.73e-97	232.40	2.38e-1064
Fronto_Parietal	-32.88	2.37e-232	99.56	1.12e-864	310.78	2.10e-1301
Default	-66.21	1.83e-565	57.94	1.36e-560	326.58	2.08e-1208
Sub_cor	-142.06	5.69e-704	-69.78	4.80e-578	145.23	1.05e-641

Table. S2. Student's *t*-test results of the system-wise modal controllability for three cases (Fig 2. b).

Comment 6. *“In the behavior prediction analysis, the author should justify why they used both autoregression and static but not sliding-window, which they show to be the best modeling for the average controllability in the first part. They considered that the graph measurement leverage more on the cognitive flexibility with a $p < 0.1$ (figure 4d and line 348); however, a p at 0.1 is a nonsignificant result. The 3D rendering seems to be just for illustration purposes as the results are only interpreted for the resting-state networks. Improvement of one model over the other should be acknowledged by the proper statistical procedure considering the number of parameters in the model (see above).”*

Response: We thank the reviewer for raising the question of justification here. First, we want to clarify that the purpose of the first part is not to determine which connectome matrix models best for controllability but to demonstrate that different static and dynamic functional connectivity would lead to distinct levels of controllability.

We considered the autoregression and static models for two reasons. First, both autoregression and sliding-window methods are to model the dynamic functional connectivity. Second, sliding-window method leads to multiple windows where the graph structure may be very noisy and makes the calculation of traditional graph measurements unreliable. For the sake of better alignment in terms of the number of features and fair comparison between control and graph theoretical measurements, we only adopt the static and auto-regression ones here. We add the explanation to the section of Prediction and Feature Importance. For the ease of review, we paste it below.

Prediction and Feature Importance

For the static and dynamic functional connectivity matrices, we calculate each regions' weighted degree, local efficiency, and participation coefficient as the graph theoretical features. On the other side, we use average controllability, modal controllability, and regional activation energy as the controllability features in the form of the control feature vector $(\varepsilon_1, \dots, \varepsilon_N, \psi_1, \dots, \psi_N, \phi_1, \dots, \phi_N)^T$ of the size $3N \times 1$ with N as the number of regions. We considered the autoregression and static models for two reasons. First, both autoregression and sliding-window methods are to model the dynamic functional connectivity. Second, sliding-window method leads to multiple windows where the graph structure may be very noisy and makes the calculation of traditional graph measurements unreliable. For the sake of better alignment in terms of the number of features and fair comparison between control and graph theoretical measurements, we only adopt the static and auto-regression ones here.

In terms of the p -values for comparing control and graph theoretical measurements used for predictive models, we did not claim that the control theoretical ones beat the graph theoretical ones. Our goal is to provide a complementary perspective to the typically used graph theoretical measurements. We agree with the reviewer that $p < 0.05$ would be a more convincing threshold to claim a significant difference for the results in Fig. 4d. However, a weakly significant p values for the case where graph

theoretical measures are better than control theoretical ones does not restrain us from using the proposed control approaches. From the perspective of hypothesis testing, one can claim that the two groups are with significantly different means when $p < 0.05$, but cannot claim that the two groups share the same means when $p > 0.05$. The latter case only indicates that we cannot very surely reject H_0 but it does not mean we can surely accept H_0 . It is the conservative philosophy of statistical tests and does not allow one to conclude reversely. Out of these reasons, we chose to honestly report the values here to display the results. The 3D surface rendering is to demonstrate how different areas contribute to the whole prediction model. We did not discuss in detail for all regions due to the constraints of paper length.

Comment 7. *“The sentence line 295-297 (“This indicates that ...”) is difficult to understand.”*

Response: In Fig. 3f-h, the x-axis is the log of the energy required to make the transition and each point represent such a transition determined by the initial and target states. The y-axis represents the ratio of additional energy when the transition is following random orders of sliding-window dynamic functional connectivity to the energy calculated with the observed order of sliding-window dynamic functional connectivity. Thus, the decreasing trend indicates that the state transition that requires less energy to drive benefits more from following the observed order of dynamic connectivity reconfiguration.

For the ease of review, we also copy the added contexts with the original statement below and hope our explanation can relieve the reviewer’s concern.

In Fig. 3f-h, the x-axis is the log of the energy required to make the transition and each point represent such a transition determined by the initial and target states. The trend then represents the relationship between the ratio of additional energy when the transition is driven by random matrix in sliding-window to the energy calculated with the observed order of dynamic functional connectivity in sliding-window. Thus, the decreasing trend indicates that the state transition that requires less energy to drive benefits more from following the observed order of dynamic connectivity reconfiguration. This indicates that the state transition that requires less energy to drive benefits more from following the observed order of dynamic connectivity reconfiguration.

Comment 8. *“The sentence on line 234 should state that the authors commonly use the eight brain networks.”*

Response: Thanks for the suggestion and we changed the sentence in the revised context. For the ease of review, we copy the revised statement below.

We adopt the eight brain networks from Yeo’s partition to compare differences in controllability for different networks^{43,49}.

Comment 9. *“The figure legends should be shortened.”*

Response: If we understand correctly, the reviewer might refer to the length of figure captions. We shortened some of the captions when possible. On the other hand, since each figure contains multiple panels, we adopted a relatively long caption so that the readers can understand the relatively complex figures without referring to the main text.

Comment 10. *“Why does the author choose the notation “6Num_of_Regions x 1”.”*

Response: Thanks for the suggestion. We change the description to “ $6N \times 1$ ” where N is the number of regions.

Reviewer 2

Summary: “In the manuscript titled “Energetic advantage in control dynamics explains dynamic reconfiguration traits during the resting state” the authors present an extension of work applying network control theory to functional-MRI by exploring the impact of replacing typically used static measures of structural connectivity as the interaction matrix in this framework with static and dynamic formulations of functional connectivity (FC). The authors show average controllability of the dynamics is higher when using sliding-window dynamic FC and modal controllability is higher when using autoregression dynamic FC. The default mode and frontal-parietal networks show a large influence on average controllability in the dynamic case and conversely the sub-cortex demonstrated large influence on modal controllability. Combining these measures with graph theory metrics is then shown to improve prediction of behavioural scores. Overall the paper is interesting and timely, both in methodology and results. After sufficient response and changes to my comments below, I recommend the paper for publication.”

Response: We thank the reviewer for the positive comments on our work regarding the methodology and results. We try our best to address the concerns below point by point.

Comment 1. *“In general the mathematics are not clearly explained as they have been applied. This forms a key part of interpreting the validity of the results in the context of the neuroscience. The method needs to be reproducible and as it stands I do not think this is the case. The mathematics is straightforward and thus a more text throughout the manuscript, but in particular the method section, describing its application to the timeseries data is needed.”*

Response: Thanks for pointing out the necessity of more clarifications on the technical details. In the revised manuscript, we described our workflow in detail. Considering the length of the main context, we attach the content in the supplement as S8: *The summary of our workflow*. For the ease of review, we paste the new text below.

S8. The summary of our workflow.

After obtaining the fMRI time series, we perform our control theoretic analyses following the pipelines below.

1. **Connectivity.** We calculate the static functional connectivity by Pearson's correlation and the dFC-Autoregression by the first-order autoregression model. The dFC-Slidingwindow is the dynamic functional connectivity sequence that uses a rectangular sliding window with a length of 60 TRs and a step size of 10 TRs.
2. **Control model.** We compute the Laplacian matrix according to Eqn. (3) and build the control model following the Eqn. (2), where \mathbf{x} is a $N \times 1$ vector that denotes the brain states, the system matrix \mathbf{A} is the Laplacian matrix from the three kinds of connectivity matrix. The input matrix \mathbf{B} consists of column vectors from the identity matrix determined by the control set, and $\mathbf{u}(t)$ denotes the system input for controlling the dynamics.
3. **Control measurements.** We calculate the average controllability (Eqn. 5) and modal controllability (Eqn. 6) for different definitions of functional connectivity including sFC-Correlation, dFC-Autoregression, and dFC-Slidingwindow. Then we compute the mean average controllability and modal controllability across all subjects and plot the spatial distribution onto the brain surface (Fig. S1). Next, we obtain the system mean of controllability for each brain network by averaging the controllability values of regions within the network and show the boxplot with errorbar (Fig. 2).
4. **Control Energy.** Given an initial state \mathbf{x}_0 and a final state \mathbf{x}_f , we compute the static control energy consumption E_s by Eqn. (4) and the dynamic energy following the method section Control with dynamic interaction matrix in the main manuscript. Each snapshot contains multiple TRs of fMRI scans. For example, the first snapshot contains 100 TRs, and we denote the activation vector at each TR as $[\mathbf{a}_1^0, \dots, \mathbf{a}_{100}^0]$. The initial state \mathbf{x}_0 is then defined as $\mathbf{x}_0 = \sum_{i=1}^{10} \mathbf{a}_{i+45} / 10$, which is the average value of the middle 10 time points. Similarly, we can define the \mathbf{x}_f from the last snapshot $[\mathbf{a}_1^f, \dots, \mathbf{a}_{100}^f]$. We use this setup to avoid that the initial and target states \mathbf{x}_0 and \mathbf{x}_f are determined by only a single time vector, which may increase the randomness. As an alternative, we can also define $\mathbf{x}_0 = \mathbf{a}_1^0$ and $\mathbf{x}_f = \mathbf{a}_{100}^f$.
5. **Control solution.** We obtain the time snapshots from the BOLD time series with window size 60 and calculate both static (whole series) and dynamic (each snapshot) function connectivity. Then we calculate the static control energy and dynamic control energy following Step 4 with varied time interval τ_s . We repeat 12 times for each subject with different τ_s choices. In Fig. 3, we set the initial and final states as the average value of the activation vector of the middle 10 BOLD time points in the first and last snapshot respectively. While in Fig. S6, we simply use the BOLD value of the first snapshot's start point as the initial state and the last snapshot's endpoint as the final state.

6. **Snapshot shuffling.** We randomly permute the snapshots 20 times and calculate the dynamic control energy with the shuffled order of snapshots. Compared with the results of the original order, we find that in most cases, disrupting the order of snapshots will increase the control energy (Fig. 3 e and Fig. S7 e). Finally, we compare the incremental ratio after shuffling the order of snapshots and the dynamic control energy following the original order of snapshots (Fig. 3 f-h and Fig. S7 f-h). If we denote the sequence of connectivity matrices as A_1, A_2, \dots, A_M , “randomly shuffle” indicates that the sequence is randomly changed to $A_{\sigma(1)}, \dots, A_{\sigma(M)}$, where σ generates a random permutation of $1, \dots, M$. The order of A_k 's makes a difference here since the time-variant interaction matrix is given by $A(t) = A_k$ for $\sum_{m=1}^{k-1} \tau_m^d < t \leq \sum_{m=1}^k \tau_m^d$, where τ_m^d is the duration of the m -th time window.
7. **Prediction of scores in the behavioral tasks.** We use three typical graph measurements include weighted degree, local efficiency, and participation coefficient. For the control measurements, we compute the average controllability, modal controllability, and regional activation energy. Each element E_i of the regional activation energy is the optimal control energy from the initial state $(0,0, \dots, 0)^T$ to the final state $e_i = (0,0 \dots, 1, \dots, 0)^T$ with the whole network as the control set. We calculate every measurement from static functional connectivity and dFC-Autoregression, and combine the same category of measurements in the two cases as the prediction feature vector. With the feature vector, we use the kernel ridge regression to predict the behavioral task scores through the 10-fold cross-validation setup. We use the Fisher-z value (statistical tests part in the main text method) for the correlation between the predicted result and the real value as the prediction performance index (Fig. 4 a and d).
8. **Complementary Power of Graph and Control Measurements.** We combine the control and graph measurements to form the new feature vector of the same length as we use in Step 7. We next predict the behavioral task scores and compare the performance through testing the difference between two correlation values.

Comment 2. “How do these functional connectivity results contrast those where the interaction matrix is a measure of structural connectivity?”

Response: We thank the reviewer for pointing out the difference between the analyses on functional and structural brain networks. The difference comes in two aspects. First, the connectivity patterns of functional and structural connectivity matrices are different. Second, the brain alters its states even in the resting state, resulting in time-varying interaction matrix. When we used the structural connectivity as the interaction matrix, we implicitly assume that the signal processes through the physical links and is analogous to the concept of effective connectivity. When we used the functional connectivity as the interaction matrix, we implicitly assume that we are investigating the random diffusion that can be characterized by the functional connectivity and driven by noise, which is viewed as a source for static functional connectivity. In terms of dynamic functional connectivity, this can only be modeled using the newly proposed

time-variant control model in this article. Based on such reasons, we conclude that it is necessary to build the control analyses on functional connectivity with some methodological adaptation. The contrast is from the intrinsic difference between structural and functional connectivity.

We also add additional discussion on the differences in interpreting the results from structural and functional connectivity in the discussion section. For the ease of review, we paste the related paragraph below.

However, the differences also exist between the two proposed networks. From the modeling perspectives, as the functional control model fits the BOLD time series directly while the structural control model studies the induced dynamics, the two frameworks are generally only applicable to their own modalities. The structural control theoretical model implicitly assumes that the signal processes through the physical links and is analogous to the concept of effective connectivity. It provides a mechanical explanation of how the underlined structure supports the executive function⁶⁴ and neural development²⁴, as well as the evolution of dynamic trajectories associated with the state transition²². Due to the limitation in modeling the dynamics from structural connectivity, it is difficult to investigate the role of the control set in driving the whole neural circuit to move across specific states when applying a fixed interaction matrix. The functional control theoretical model implicitly assumes that the random diffusion of brain signal is characterized by the static functional connectivity and the non-diffusive part can be inferred from the time-variant functional connectivity. Thus the proposed control theoretical analyses based on both static and dynamic functional connectivity potentially bridge the gap by analyzing the time-series directly rather than inferring from the structure²⁷. This is also adopted in a recent work⁵⁸ but with effective connectivity. The functional control framework allows future application of intervening the neural circuits via certain nodes for psychiatric medication⁷⁵.

Comment 3. “Line 252: “Energy Efficiency Explains the Dynamic Reconfiguration of Functional Connectivity.” The word ‘Explained’ is too strong here

Response: Thanks for pointing out the inaccuracy in expression. We changed it to “Control Theory Illustrates the Energy Efficiency in the Dynamic Reconfiguration of Functional Connectivity”.

Comment 4. “Fig 3: This caption is insufficient. Explain what is plotted. Discussion of results is the purpose of main text.”

Response: Thanks for the suggestion. We have added additional explanation of the plotted dots in the revised Fig 3. Specially, we added the following text to help explain the axis labels.

The y-axis of (f)-(h) represents the log of the relative energy inflation when changing the connectivity matrices from the observed order to the random order. (a) We successively set Default Mode Network (DM), the combination (CN) of Default Mode, Dorsal Attention (DA), Salience (SA),

and Fronto-parietal networks (FP), or the whole network (All) as the control set. The T-tests measure their significance vs. zero.

Comment 5. “Line 256: What is randomly shuffled?”

Response: Thanks for pointing out the potential confusion here. For the case of dFC-Slidingwindow, if we denote the sequence of connectivity matrices as A_1, A_2, \dots, A_M , “in a randomly shuffled order” means that the sequence is randomly changed to $A_{\sigma(1)}, \dots, A_{\sigma(M)}$, where σ generates a random permutation of $1, \dots, M$. The order of A_k 's makes a difference here since the time-variant interaction matrix is given by $A(t) = A_k$ for $\sum_{m=1}^{k-1} \tau_m^d < t \leq \sum_{m=1}^k \tau_m^d$, where τ_m^d is the duration of the m -th time window.

We add this explanation to the supplement S8 *The summary of our workflow* as well for a better reference. For the ease of review, we paste the text below.

Snapshot shuffling. We randomly permute the snapshots 20 times and calculate the dynamic control energy with the shuffled order of snapshots. Compared with the results of the original order, we find that in most cases, disrupting the order of snapshots will increase the control energy (Fig. 3 e and Fig. S7 e). Finally, we compare the incremental ratio after shuffling the order of snapshots and the dynamic control energy following the original order of snapshots (Fig. 3 f-h and Fig. S7 f-h). If we denote the sequence of connectivity matrices as A_1, A_2, \dots, A_M , “randomly shuffle” indicates that the sequence is randomly changed to $A_{\sigma(1)}, \dots, A_{\sigma(M)}$, where σ generates a random permutation of $1, \dots, M$. The order of A_k 's makes a difference here since the time-variant interaction matrix is given by $A(t) = A_k$ for $\sum_{m=1}^{k-1} \tau_m^d < t \leq \sum_{m=1}^k \tau_m^d$, where τ_m^d is the duration of the m -th time window.

Comment 6. “Line 272: *significantly”

Response: Thanks for pointing out the grammar fault. We have corrected it in the revision.

Comment 7. “Line 586: Do you actually evaluate this integral over $t=0$ -inf or do you impose a time horizon? Please explain exactly how you've implemented the mathematics.”

Response: Thanks for pointing out the potential confusion here. In practice, we did not solve W_i with the integration. There are two typical approaches. One is to solve the Lyapunov equation $AW_i + W_iA^T + BB^T = 0$ for W_i and the other is to follow the formula $\psi_i = \text{Trace}[(I - A^2)^{-1}]$ derived in the supplement of Gu et al., Nat. Comm. 2015. In the revised manuscript, we explicitly state how to solve W_i given A and B.

Comment 8. “Line 606: “We set the initial state x_0 and the final state x_f as the average value of the middle 10 time points in the initial and final snapshot” What does

“this mean?”

Response: Thanks for pointing out the potential confusion here. Each snapshot contains multiple TRs of fMRI scans. For example, the first snapshot contains 100 TRs and we denote the activation vector at each TR as $[\mathbf{a}_1^0, \dots, \mathbf{a}_{100}^0]$. The initial state \mathbf{x}_0 is then defined as $\mathbf{x}_0 = \sum_{i=1}^{10} \mathbf{a}_{i+45}/10$, which is the average value of the middle 10 time points. Similarly, we can define the \mathbf{x}_f from the last snapshot $[\mathbf{a}_1^f, \dots, \mathbf{a}_{100}^f]$. We use this setup to avoid the situation in which the initial and target states \mathbf{x}_0 and \mathbf{x}_f are determined by only a single time vector, which may increase the randomness. As an alternative, we can also define $\mathbf{x}_0 = \mathbf{a}_1^0$ and $\mathbf{x}_f = \mathbf{a}_{100}^f$. We add this explanation to the supplement S8 *The summary of our workflow* as well for a better reference. For the ease of review, we paste the text below.

Control Energy. Given an initial state \mathbf{x}_0 and a final state \mathbf{x}_f , we compute the static control energy consumption E_s by Eqn. (4) and the dynamic energy following the method section *Control with dynamic interaction matrix* in the main manuscript. Each snapshot contains multiple TRs of fMRI scans. For example, the first snapshot contains 100 TRs, and we denote the activation vector at each TR as $[\mathbf{a}_1^0, \dots, \mathbf{a}_{100}^0]$. The initial state \mathbf{x}_0 is then defined as $\mathbf{x}_0 = \sum_{i=1}^{10} \mathbf{a}_{i+45}/10$, which is the average value of the middle 10 time points. Similarly, we can define the \mathbf{x}_f from the last snapshot $[\mathbf{a}_1^f, \dots, \mathbf{a}_{100}^f]$. We use this setup to avoid that the initial and target states \mathbf{x}_0 and \mathbf{x}_f are determined by only a single time vector, which may increase the randomness. As an alternative, we can also define $\mathbf{x}_0 = \mathbf{a}_1^0$ and $\mathbf{x}_f = \mathbf{a}_{100}^f$.

We performed the similar analyses following this setup and added the corresponding results in the supplement as Fig. S7. The results almost remain the same as before. For the ease of review, we copy the related figure and context below.

Fig. S7. Energy Efficiency Explains the Dynamic Reconfiguration of Functional Connectivity. As an alternative to the results shown in Fig. 3, we change the initial state to the first time point BOLD vector and the final state to the last time point BOLD vector of the stime series. (a-d) Dynamic control paradigm is energetically more efficient than that of static control no matter

we set the control set as Default Mode Network (DM), the combination of Default Mode, Dorsal Attention (DA), Salience (SA), and Fronto-parietal networks (FP), or the whole network (All). (e) The driving energy cost will increase when the order of snapshots is disrupted. (f-h) The relative increase of log-energy after shuffling the snapshot order is negatively related to the log-energy of dynamic control consummation.

Comment 9. *“Line 606: The idea behind this paragraph is good, i.e., walk through the mathematics as you have applied it, but the descriptions given here are muddled and full of jargon not defined. A supplementary figure detailing the methods application to the timeseries would be helpful.”*

Response: Thanks for the suggestion. We respond to the request in Comment 1 & 9 together and added a new section in the supplement to address this shortcoming. Please refer to the response above for the content.

Comment 10. *“Fig. 4d *participation coefficient.”*

Response: Thanks for pointing out the grammar fault. We correct it in the revision.

Reviewer 3

Summary: In this work, Deng and colleagues give an analysis tour-de-force to improve our current understanding of brain networks and how they are related to behavior. This is a great example on how to use large-scale, open-source, neuroimaging databases (in this case HCP, resting-state fMRI) to propose both methodological advances that have testable theoretical implications. I have a few concerns, though, that preclude the acceptance of this paper as it is:

Response: We appreciate the reviewer for the acknowledgement of our contribution to developing novel tools in network neuroscience. Also, we thank the reviewer for raising questions on our article to help us improve its quality. We try our best to address the questions one by one below.

Comment 1. *“For the results provided for Fig 1, the data is described as “the average controllability of the dynamics induced from dFCslidingwindow is higher than that of the sFC-Correlation and dFC-Autoregression (Fig. 2c) except for the subcortical areas” in the results and “In general, the average controllability of the brain system parameterized by dFCslidingwindow is higher than the systems parameterized by sFC-Correlation” in the Figure, but no statistics are provided. If the authors want to keep this analysis as it is, then a better explanation of why no statistical comparison between methods is warranted here.”*

Response: Thanks for pointing out the lack of statistics in Fig. 2. As suggested by Reviewer 1, we modified Fig. 2 with the surface panels removed. In addition, we adopt

your suggestions and add the significance level of the t-tests for each system separately. For the ease of review, we also paste the new Fig. 2 below and supplementary statistics below.

Revised Fig. 2:

Fig. 2. Spatial Distribution of Functional Controllability. We show the distribution of mean (a) average and (b) modal controllability across the different functional brain networks. The mean average control values (Mean ac-Value) and mean modal control values (Mean mc-Value) are calculated across all subjects. The system mean is calculated across the within-system regions for all subjects. The 8 brain networks are the commonly used Yeo’s partition when investigating functional networks^{44,50}.

Additional Results in the supplement:

	Static FC VS. dFC-Autoregression		Static FC VS. dFC-Slidingwindow		dFC-Autoregression VS. dFC-Slidingwindow	
	T value	p-value	T value	p-value	T value	p-value
	Visual	34.66	5.85e-230	-38.90	2.64e-287	-75.53
Somatomotor	53.37	3.41e-326	-13.02	4.93e-62	-83.00	1.64e-402
Dors_Attn	59.38	3.27e-411	-45.68	4.01e-340	-97.43	2.74e-474
Saliency	105.97	5.95e-573	-8.52	3.75e-23	-96.10	4.11e-478
Limbic	94.90	9.42e-517	-5.34	5.36e-10	-82.89	3.54e-423
Fronto_Parietal	44.42	1.46e-300	-106.34	6.25e-615	-143.28	7.96e-633
Default	72.24	1.22e-496	-64.34	2.17e-429	-115.15	3.86e-537
Sub_cor	96.47	2.24e-504	48.02	2.85e-411	-43.43	1.86e-217

Table. S1. Student's t-test results of the system-wise average controllability from three cases (Fig 2. a).

	Static FC VS. dFC-Autoregression		Static FC VS. dFC-Slidingwindow		dFC-Autoregression VS. dFC-Slidingwindow	
	T value	p-value	T value	p-value	T value	p-value
	Visual	-30.79	2.42e-225	32.54	4.26e-294	195.78
Somatomotor	-61.66	7.92e-438	3.83	2.58e-7	194.60	3.92e-760
Dors_Attn	-59.62	4.54e-492	38.27	1.47e-353	292.99	2.29e-1051
Saliency	-121.96	4.04e-744	-10.69	2.13e-41	296.54	1.17e-1138
Limbic	-115.88	5.26e-728	-17.33	5.73e-97	232.40	2.38e-1064
Fronto_Parietal	-32.88	2.37e-232	99.56	1.12e-864	310.78	2.10e-1301
Default	-66.21	1.83e-565	57.94	1.36e-560	326.58	2.08e-1208
Sub_cor	-142.06	5.69e-704	-69.78	4.80e-578	145.23	1.05e-641

Table. S2. Student's t-test results of the system-wise modal controllability from three cases (Fig 2. b).

Comment 2. "The authors say "Here we choose the fluid intelligence and cognitive flexibility as two representative tests considering their comprehensive involvement of multiple brain system during execution". As seen in Figure 5a, there are many other behavioral scores that could have been selected as they also involve multiple brain systems during execution (sustained attention, working memory, as an example). The authors need to justify better why cog flexibility and fluid intelligence were chosen or

why performance is worse for other behavioral scores that should also depend on multiple brain systems.”

Response: Thanks for pointing out the lack of explanation of the choice here. We chose the fluid intelligence and cognitive flexibility as the representative tests as they displayed the biggest difference when predicting with graphical or control measurements respectively. In terms of why they overperforms other behavioral scores, admittedly we do not have very strong reasons. We think it might be because they are integral scores of many factors and are thus more reliable than other behavioral scores.

Comment 3. *““We choose the one-tailed set here as we are interested in whether the combination of control and graph measurements outperforms each separately rather than whether they are different from each other”. This needs a better justification, taking into account that two-tailed t-tests are used in other analysis.”*

Response: Thanks for pointing out the lack of clarification here. The two-tailed set is to test whether the two groups are different while the one-tailed set is to test whether one is greater than the other. In our case, we are interested in whether the combination is better thus we test versus the non-hypothesis “combo is worse than single”, which is the one-tailed set we used.

Comment 4. *“Fig3. The p-values in the figure are difficult to see.”*

Response: Thanks for the suggestion. We have increased the font and optimized the location in the revised manuscript.

Comment 5. *““This might explain why the two types of measurements can complements each other” should be complement.”*

Response: Thanks for pointing out the grammar error. We have corrected it in the revised manuscript.

Reviewers' comments:

Reviewer #1 (Remarks to the Author):

Thanks to the author for this documented rebuttal.

I fiercely disagree with the author when they advocated that quantitative results are unimportant, a least not enough to be in the abstract. Results are not "purely technical improvements"; it is the aim of our scientific work. It is, even more bothering that they have such results in their analysis that could make a great article.

I agree with most of the answers, but some points need clarifications.

Concerning the analysis shown in Figure 3. The authors agree that their analysis is artificial because they A) mixed intra- and inter-subject variability and B) that the number of tests involved in the intrasubject variability is arbitrarily set at a high number; This latter leads to the incredible low p-values. The proposed figure S8 should solve this question; however, I doubt that inter-subject variability with the variability shown on the figure can lead to $p=1.831e-30$. I think that the requested Gaussian assumptions are not met. Nevertheless, Figure S8 should be in the paper in place of Figure 3, as the analysis is sounder. Furthermore, for each test (in the article), you should give the number of samples as customary when reporting statistics.

Figure 4 shows promising results, but I think the legend describes an effect opposite to what is shown in figure (part d). It seems to me that it is the graph-theoretical measurement predicts better than the model theoretical measurement.

Figure 5 is not conclusive as the statistical tests of the improvements of one method to the other are missing.

In the preference in control strategy, the cited reference 50 should be 49.

Reviewer #2 (Remarks to the Author):

The authors have adequately addressed my concerns on the previous manuscript and made changes where appropriate. I recommend this article for publication.

Reviewer #3 (Remarks to the Author):

The authors have provided appropriate responses to all my concerns.

COMMSBIO-21-1464-B

Control Theory Illustrates the Energy Efficiency in the Dynamic Reconfiguration of Functional Connectivity

Shikuang Deng, Jingwei Li, B.T. Yeo, Shi Gu

We thank all the reviewers for their efficient feedback on our revised manuscript. Based on the further requests on clarification, we have carefully addressed each comment on by one and revise the manuscript respectively. We believe that the manuscript has been greatly improved after these two rounds of reviews.

Reviewer 1

We really appreciate the insightful and careful comments on both the organization and technical details. The constructional suggestions help us improve the soundness and readability from its original version. In the following context, we will respond to the comment piece by piece.

Comment 1. *“I fiercely disagree with the author when they advocated that quantitative results are unimportant, a least not enough to be in the abstract. Results are not “purely technical improvements”; it is the aim of our scientific work. It is, even more bothering that they have such results in their analysis that could make a great article.”*

Response: We adopt the reviewer's suggestion that the abstract can be revised to provide additional quantitative results. We rework the abstract as indicated in the revision and include it below for convenience of review.

The brain's functional connectivity fluctuates over time instead of remaining steady in a stationary mode even during the resting state. This fluctuation establishes the dynamical functional connectivity that transitions in a non-random order between multiple modes. Yet it remains unexplored how the transition facilitates the entire brain network as a dynamical system and what utility this mechanism for dynamic reconfiguration can bring over the widely used graph theoretical measurements? To address these questions, we propose to conduct an energetic analysis of functional brain networks using resting-state fMRI and behavioral measurements from the Human Connectome Project. Through comparing the state transition energy under distinct adjacent matrices, we justify that dynamic functional connectivity leads to 60% less energy cost to support the resting state dynamics than static connectivity when driving the transition through default mode network. Moreover, we demonstrate that combining graph theoretical measurements and our energy-based control measurements as the feature vector can provide complementary prediction power for the behavioral scores (Combination vs. Control: $t = 9.41$, $p = 1.64e-13$; Combination vs. Graph: $t = 4.92$, $p = 3.81e-6$). Our approach integrates statistical inference and dynamical system inspection towards understanding brain networks.

Comment 2. *“Concerning the analysis shown in Figure 3. The authors agree that their analysis is artificial because they A) mixed intra- and inter-subject variability and B) that the number of tests involved in the intrasubject variability is arbitrarily set at a high number; This latter leads to the incredible low p-values. The proposed figure S8 should solve this question; however, I doubt that inter-subject variability with the variability shown on the figure can lead to $p=1.831e-30$. I think that the requested Gaussian assumptions are not met. Nevertheless, Figure S8 should be in the paper in place of Figure 3, as the analysis is sounder. Furthermore, for each test (in the article), you should give the number of samples as customary when reporting statistics.”*

Response: We replace panel (a) and (e)-(h) in Figure 3 with Figure S8 as suggested. Considering that we have 865 subjects, a very small p-value corresponding to $r \approx 0.36$ is reasonable. In terms of the Gaussian assumptions here, it is possible that we calculate the p-value with permutation test. But considering the high significance here, this alternative approach won't change the conclusion thus we didn't do that for simplicity. Finally, we thank the reviewer's advice on adding the number of samples as customary when reporting statistics. It has been added in the revised manuscript.

Comment 3. *“Figure 4 shows promising results, but I think the legend describes an effect opposite to what is shown in figure (part d). It seems to me that it is the graph-theoretical measurement predicts better than the model theoretical measurement.”*

Response: Thanks for pointing out the error in the legend. We have corrected it in the revision.

Comment 4. *“Figure 5 is not conclusive as the statistical tests of the improvements of one method to the other are missing.”*

Response: The explanations are already in the context with statistics located on lines 401-409. For the ease of review, we paste it below.

The same improvement of using combined features exhibits on the three subgroups of tasks as well, no matter the task score can be predicted by control measurements (Fig. 5b, Combination vs. Control: $t = 4.07$, $p = 3.29e-4$; Combination vs. Graph: $t = 4.84$, $p = 5.63e-5$), graph measurements (Fig. 5c, Combination vs. Control: $t = 5.49$, $p = 6.04e-6$; Combination vs. Graph: $t = 1.85$, $p = 3.81e-2$), or their combination (Fig. 5d, Combination vs. Control: $t = 7.79$, $p = 2.31e-9$; Combination vs. Graph:

t = 5.74, p = 9.35e-7). The prediction improvement in Fig. 5b-d is calculated as the z-values of the difference of correlations between the predicted and observed scores.

Comment 5. *“In the preference in control strategy, the cited reference 50 should be 49.”*

Response: Thanks for pointing out the reference error due to the format of citation. We have corrected it in the revised manuscript.

REVIEWERS' COMMENTS:

Reviewer #1 (Remarks to the Author):

The authors have provided appropriate responses to my concerns.